# Response to Cadmium Toxicity: Orchestration of Polyamines and microRNAs in Maize Plant

**DOI:** 10.3390/plants12101991

**Published:** 2023-05-15

**Authors:** Seyedeh Batool Hassani, Mojgan Latifi, Sasan Aliniaeifard, Shabnam Sohrabi Bonab, Neda Nasiri Almanghadim, Sara Jafari, Elham Mohebbifar, Anahita Ahangir, Maryam Seifikalhor, Hassan Rezadoost, Massimo Bosacchi, Anshu Rastogi, Françoise Bernard

**Affiliations:** 1Department of Plant Sciences and Biotechnology, Faculty of Life Sciences and Biotechnology, Shahid Beheshti University, Tehran 19839-69411, Iran; b_hassani@sbu.ac.ir (S.B.H.);; 2Photosynthesis Laboratory, Department of Horticulture, College of Agricultural Technology (Aburaihan), University of Tehran, Tehran 33916-53755, Iran; 3Parcham Company, Pakdasht, Tehran 33916-53755, Iran; 4Department of Phytochemistry, Medicinal Plants and Drugs Research Institute, Shahid Beheshti University, Tehran 19839-69411, Iran; 5Park at the Danforth Plant Science Center, KWS Gateway Research Center, LLC, BRDG, Saint Louis, MO 95618, USA; 6Laboratory of Bioclimatology, Department of Ecology and Environmental Protection, Faculty of Environmental Engineering and Mechanical Engineering, Poznan University of Life Sciences, Piątkowska 94, 60-649 Poznań, Poland

**Keywords:** cadmium stress, microRNAs, photosynthesis, putrescine, *Zea mays* L.

## Abstract

Cadmium (Cd) is a heavy metal that is widely contaminating the environment due to its uses in industries as corrosive reagents, paints, batteries, etc. Cd can easily be absorbed through plant roots and may have serious negative impacts on plant growth. To investigate the mechanisms utilized by plants to cope with Cd toxicity, an experiment was conducted on maize seedlings. We observed that the plant growth and photosynthetic mechanism were negatively influenced during 20 days of Cd stress. The expression levels of ornithine decarboxylase (*ORDC*) increased in the six seedlings under Cd exposure compared to the control. However, Cd toxicity led to an increase in putrescine (Put) content only on day 15 when compared to the control plants. In fact, with the exception of day 15, the increases in the *ORDC* transcript levels did not show a direct correlation with the observed increases in Put content. Spermidine and Spermine levels were reduced on day 6 by Cd application, which was parallel with suppressed Spermidine synthase gene. However, an increase in Spermidine and Spermine levels was observed on day 12 along with a significant elevation in Spermidine synthase expression. On day 6, Cd was observed to start accumulating in the root with an increase in the expression of microRNA 528; while on day 15, Cd started to be observed in the shoot part with an increase in microRNA 390 and microRNA 168. These results imply that different miRNAs may regulate polyamines (PAs) in maize under Cd toxicity, suggesting a plant-derived strategy to commit a PAs/miRNA-regulated mechanism/s in different developmental stages (time points) in response to Cd exposure.

## 1. Introduction

Cadmium (Cd) is one of the most persistent trace metals and a hazardous environmental pollutant that causes diverse biochemical and physiological malfunctions in plants [1]. Cd accumulates in the soil as a byproduct of industrial and municipal wastewater, and as a result of the unnecessary application of insecticides and phosphorus-based fertilizers [2]. Since Cd is not a necessary nutrient for the vital processes of plant metabolism and growth, its concentration in the soil should be low enough to prevent the risk of contamination, especially considering its rapid mobility in the soil and durability in the plant—animal and human food chain [3]. The biological interaction of Cd with the sulfhydryl groups found in the structure of proteins and enzymes, which interfere with their activity, causes Cd toxicity in plants. Furthermore, comparable divalent cations can readily replace Cd in the active sites of antioxidant proteins, creating an imbalance in cellular reactive oxygen species (ROS). Alternatively, Cd induces phytochelatin biosynthesis and serves non-enzymatic antioxidant glutathione (GSH) depletion, which indirectly favors an ROS boost [4,5,6].

Although unfavorable conditions, such as excessive Cd exposure, negatively impact plant growth and development, plants have evolved sophisticated physiological mechanisms to survive under such conditions [7,8,9,10]. To combat Cd-induced malfunctions, plants have evolved enzymatic antioxidants, including superoxide dismutase (SOD), catalase (CAT), and other peroxidases. Besides, as well-known plant bio-stimulant metabolites, PAs carry out the critical stress-management role in plant defense strategy [10]. PAs such as Put, spermidine (Spd), and spermine (Spm) interact with negatively charged macromolecules and regulate their functions. They may also act as compatible osmolytes or provide antimicrobial activity against plant pathogens. In addition, PAs stimulate the antioxidant defense system and are considered key factors in reducing oxidative stress according to their ability to scavenge ROS and free radicals [11,12,13]. Benavides and coworkers (2018) demonstrated the elimination of the deleterious effect of Cd by PA exogenously applied to wheat (*Triticum aestivum* L.) plants [14]. The protective function of PA against Cd-induced oxidative damage in sunflowers has also been validated [15].

Previous reports indicate that the expression of several genes participating in the PA synthesis pathway increases when exposed to one or more abiotic stresses. Putrescine is the first PA accumulated in cells under abiotic stresses. It is noteworthy that increasing the concentration of Put leads to the stimulation of the enzymes responsible for the conversion of Put into Spd and Spm, through the self-regulation process [16]. For instance, exposure of the *Potamogeton crispus* leaves to Cd raised the level of Put while decreasing Spm and Spd [17]. The overexpression of heterologous *Spd synthase* (*SPDS*), *S-Adenosylmethionine decarboxylase* (*SAMDC*), and *ORDC* in different plant species, such as rice, tobacco, and tomato, caused tolerance to a wide range of stress conditions [18]; an increase in tolerance is always related to an increase in Put or Spd and Spm. In *Potamogeton crispus*, the effects of Cd exposure on structure, accumulation, and PA reduction were investigated. Cd treatment increased Put while decreasing Spm and Spd contents, which decreased the total ratio of free Spm and Spd compared to Put in the leaves. Conjugated PAs changed in the same way and with the same pattern, while bonded PAs showed a different pattern. The content of band Put increased with the increase of Cd concentration up to 50 µM and then decreased. The content of the Spm and Spd bands decreased to a lesser extent. It has been shown that PAs and their particular forms can play an important role in the adaptation mechanism of *Potamogeton crispus* under Cd stress [17]. In another study, the effects of different concentrations of Cd and copper (Cu) on the growth and metabolism of PAs in sunflower stems were investigated. At 1 mM concentration of Cd and Cu, Put content increased by 270% under the influence of Cd ions, and 260% under the influence of Cu ions. Spd changed at 1 mM Cd concentration, while Spm decreased after seed germination and then increased at 1 mM Cd or Cu concentration [19]. 

However, the precise physiological and molecular mechanisms underlying PA metabolism and catabolism, as plant stress-related metabolites, remain under debate [9,14,20,21]. 

Evidence collected from studies investigating plant tolerance mechanisms implicates thousands of genes that are transcriptionally/translationally altered in response to biotic and abiotic stresses [7,8,22,23,24]. Deciphering the mechanisms underlying the fine-tuning of stress-related genes in plants has led to the identification of micro RNAs (miRNAs) as potential targets to regulate gene expression under stress events. 

Recent discoveries have shown that miRNAs regulate gene expression in plants when they are subjected to environmental stresses. These small, non-coding RNAs are abundant in higher organisms and play important roles in regulating different gene expressions. To date, 1600 miRNAs have been identified in plants and animals with critical roles in regulating gene expression and various cellular processes. The miRNAs inhibit gene translation by affecting the target mRNA, either by RNA degradation or by blocking RNA translation outwork [25]. 

Stress-responsive miRNAs have been associated with both biotic [26,27,28] and abiotic stresses such as metal toxicity [29,30,31,32,33]. MiRNAs responses to Cd in *Medicago truncatula* have been identified by Zhou et al. [34]. More studies also reported altered the expression of miRNAs in *Brassica napus* [35,36], rice [37,38], and *Arabidopsis thaliana* [31] after Cd exposure [39]. However, our understanding of miRNA-mediated responses to Cd toxicity in plants is still incomplete. The differential expression of miRNAs under metal toxicity is consistent with a complex regulatory role [20,40,41]. This implies that, in spite of the generic role defined for miRNAs against stress events, different stimuli can trigger specific miRNA-mediated responses. Notably, among studies performed on miRNA/Cd-related responses, the roles of these regulatory elements in the plant developmental stages are lacking. As plants diversely respond to environmental stresses during growth and development [42,43,44], it is critical to understand the regulatory networks underlying dynamic plant responses modulated by miRNAs. In this study, we attempted to identify the cellular mechanisms involved in the development of Cd resistance in maize plants. Given that the plant response to the environment is a complex and dynamic process, the present study was conducted over different time points starting from 3 to 20 days post Cd exposure. With the underlying notion that the miRNAs and PAs have been introduced as cellular regulators, the role of miRNAs and Pas metabolism associated with physio/morpho-logical responses has been investigated. Understanding the miRNA mechanism of regulation against abiotic stress will help to manipulate susceptible crop plants and increase agricultural productivity shortly.

## 2. Results

Under control conditions, shoot and root length increased as expected over the course of 20 days. Cd caused negative effects on both root and shoot lengths when compared to the control plants at each time-point (Figure 1a,b). Similar results were obtained in the leaf area (LA); the LA in Cd-exposed plants was half of that in control at all time points (Figure 1c).

Cd content in the root increased in a time-dependent manner until day 12, then dropped dramatically (70%) between days 15 and 20 (Figure 2a). The Cd content in the shoot increased in a time-dependent manner until day 15, followed by an 18% decrease at day 20 (Figure 2b). The Cd translocation factor represents a significant accumulative movement of Cd from the root to shoot in a time-dependent manner (Figure 2c). 

### 2.1. Inhibitory Effect of Cd on Photosynthesis Functionality of Maize Plant

Photosynthetic performance—including Fv/Fm, NPQ and PiAbs—was significantly reduced (nearly 15%) as a result of the Cd application (Figure 3a). ABS/RC increased by 15% in plants exposed to Cd, however, a nearly 10% reduction was observed in ET_0_/RC and TR_0_/RC and DI_0_/RC by Cd application (Figure 3b). 

### 2.2. Expression Pattern of Peroxisomal and Apoplastic PAOs under Cd Toxicity

An induction of *PAO1* expression was observed under control conditions from day 6 to 20. However, Cd exposure suppressed *PAO1* gene expression in comparison with the controls, increasing *PAO1* expression was observed in stress conditions until day 15, followed by a significant decrease (10%) on day 20 (Figure 4a). 

*PAO2* expression remained steady under control conditions over 20 days of growth. In contrast, under stress conditions, it abundantly (ten times) increased on day 12, followed by a sharp decrease (10 times) on day 15. A twofold induction increase was observed further on day 20 by Cd application (Figure 4b). 

*PAO3* expression remained constant under control conditions until day 12 and further showed a doubling of the induction rate on days 15 and 20. Cd exposure resulted in a significant increase in *PAO3* expression on days 3 and 12. However it led to a decrease in *PAO3* gene expression on day 6, and an increase was recorded on day 12. Cd application reduced *PAO3* expression to the minimum level on days 3, 15 and 20 (Figure 4c). 

*PAO4* expression slightly increased under control conditions until day 15 and further abundantly (nearly 6 times) expressed on day 20. A similar pattern was observed with Cd shock; however, on day 20, *PAO4* expression was significantly lower than the control condition (Figure 4d).

Under control conditions, *PAO5* expression was slightly elevated from day 3 to 12 followed by 2 and 7 times increase on days 15 and 20, respectively. Under Cd stress, *PAO5* gene expression showed a 3 fold increase on days 6 and 12 compared to its expression on day 3, followed by a sharp increase (5 times) on days 15 and 20 (Figure 4e).

In contrast with other *PAO* genes, under control conditions, *PAO6* was significantly reduced till day 12; however, an increase was detected at 15 and 20 days afterwards. In the opposite manner, Cd exposure suppressed *PAO6* expression on day 3 and thereafter an increase (6 times) until day 12 was observed. On day 15 under Cd shock, *PAO6* expression reduced 3 times, followed by a sharp increase (5 times) on day 20 (Figure 4f).

### 2.3. PAs Levels in Response to Cd Toxicity

Total Put did not change in control plants during the 20 days; however, in Cd-exposed plants, the first induction was observed on day 6 and the maximum amount was recorded on day 15. In the rest dates, the total Put remained at the same value (Figure 5a). 

The highest total Spd was detected in plants grown in both stressed and non-stressed conditions on day 3; however, in both conditions total Spd content reduced up to 50% on day 6 following a dramatic increase on day 12. A further 40% reduction was observed in total Spd content in plants grown in both conditions (Figure 5b). By Cd contamination, Spd and Spm content increased on day 12. However, day 15 showed the highest Put content; nevertheless, Put content was higher than the other PAs during 20 days (Figure 5).

Although total Spm was at the maximum value on day 3 in control plants, its amount was significantly lower than the total Put on the same date. Moreover, a significant reduction (nearly 40%) was observed in a time-dependent manner until day 15. The total Spm level in Cd exposed plants was similar to its value in control plants on day 3. On day 6, total Spm was reduced (three times) followed by a significant induction on day 12. Although a further reducing trend was observed until the Spm value reached 0 on day 20 (Figure 5c).

### 2.4. Differential Expression of PAs Biosynthesis-Related Genes by Cd Exposure

*SPDS* gene expression increased in a time-dependent manner under control conditions until day 15. However, its expression was reduced on day 20. In plants grown under Cd shock, *SPDS* expression was reduced through day 6 and showed significant expression on day 12; however, its expression level was still lower than control plants. A further dramatic reduction was recorded in *SPDS* on days 15 and 20 (Figure 6a). *ORDC* gene expression was significantly increased on day 6 in control plants. However, a dramatic reduction was recorded on day 12, followed by an increase on day 15. In plants treated with Cd, a significant increase (150 times) was observed on day 6, however, a further dramatic reduction was detected (70 times) on day 12. Later on day 15, a two times increase was followed by a reduced expression on day 20 (Figure 6b). *ARGDC* expression remained steady in control plants at all time points except for day 12, which showed a significant induction. By contrast, in plants treated with Cd, *ARGDC* expression was abundantly high on day 3 and reduced in a time-dependent manner afterwards (Figure 6c). 

### 2.5. Regulated Expression Pattern of miRNAs by Cd Application 

The miR390 expression was maximum on day 3 in control plants. However, a dramatic reduction was recorded on days 6 and 12 for miR390 expression, and a significant increase was observed afterwards on days 15 and 20. In Cd-treated plants, miR390 expression remained steady until day 6 and was reduced to a minimum value on day 12; however, a significant expression was recorded afterwards on days 15 and 20 (Figure 7a).

For control plants, miR528 expression levels were steady at all time points. By contrast, in Cd-exposed plants, a significant induction of miR528 was observed from day 3 to day 6; however, its expression level was reduced later on days 12, 15 and 20 (Figure 7b).

Although the minimum expression level of miR168 was recorded on day 3 in control plants, a significant increase (nearly 60 times) was observed on day 6, nevertheless, its expression level was reduced to the same value as on day 3 and day 12. Later, a significant increase (17 times) was observed in the miR168 expression level on day 15, followed by two-times reduction on day 20 that increased in a time-dependent manner afterwards until day 20 (Figure 7c).

## 3. Discussion

There is an abundance of data focusing on the negative effects of Cd on plant growth and development [45,46,47]. However, in the present work, when plants were exposed to Cd contamination, shoot length increased until day 15 followed by a significant cessation by day 20. Competing for absorption sites is typically one of the mechanisms underlying Cd assimilation by plant roots [48]. It has been shown that the electrochemical potential difference between Cd activity in the cytosol and in the root’s apoplasts controls cadmium absorption across the plasma membrane of root cells [45]. Toxic metal deposition in the cell wall, altered toxic compounds, sequestration/accumulation, and changes in the cell wall plasma membrane complex have all been implicated as methods of resistance to microbial Cd [49]. Nevertheless, it has been proved that hyper-accumulating plants harbour sequestration, detoxification, and storing strategies to promote resistance mechanisms against trace metal ions, such as Cd [3]. In this regard, cuticles, epidermis, and trichomes are tissues for which the majority of metal detoxification and storage occurs, where the photosynthesis activity is the least [50]. This can imply that, during the evolutionary trajectory of heavy metal resistance, plants have gained protection mechanisms to commit to beneficial functions for plants under toxic minerals such as Cd. 

The Cd translocation factor correlates this cessation with Cd levels reaching their highest point on day 20. One of the well-known mechanisms developed by plants is the engagement of PAOs under various stress-related conditions. Our previous study showed that maize plants exert mechanisms underlying PAs-related responses to cope with Cd shock [10]. PAOs play important roles in programmed cell death events by mediating H_2_O_2_ signaling, leading to Spd, Spm, and Spm oxidation. PAOs have localized in cells according to their structure. For example, ZmPAO2, 3, 4 and 6 were predicted to contain peroxisomal-targeting signals in their C-terminal, which direct them to be localized in the peroxisome. Whereas ZmPAO1 and PAO5 have been detected in the apoplast. Although most intracellular PA catabolism occurs in peroxisomes [51], the existence of a different type of PAOs suggests a tight regulatory mechanism between enzymes that preserve the PA cellular content at the optimum level [52].

PAO enzymes are involved in a plethora of developmental procedures and exhibit different subcellular localization, substrate specificity, and functional diversity [12]. Wang and Liu found that CsPAO3 may potentially catalyze the PA back conversion in plants, whereas CsPAO4 catalyzes Spd and Spm as substrates for terminal catabolism [53]. Alternatively, elevating the CO_2_ in apple fruit in low-temperature/low-O_2_ storage has been observed by Brikis et al. by the expression of *MdPAO2* [54]. Nevertheless, spatial and temporal expression of *PAOs* is regulated by environmental and endogenous stimuli. Due to the presence of a simultaneous breakdown route within the cell, which generates H_2_O_2_, the quantity of Put, Spd, and Spm in cells frequently fluctuates. This H_2_O_2_, which is created as a result of the catabolism of PAs, contributes to oxidative stress and is crucial for the lignification of the cell wall, which shields the plant from the negative effects of stress stimuli [16]. The role of different PAOs in various stress conditions has been shown previously; for instance, AtPAO2 and AtPAO5 up-regulation were reported in saline stress in Arabidopsis thaliana plant [55]. In a different study of a double mutant for peroxisomal PAOs, atpao2-atpao4, were found to be sensitive to drought stress [56]. In the current study, PAOs expression dynamically changed at different time points. Similar dynamics were shown in the expression of PAO2 and PAO3 until day 12 after exposure to Cd, while on day 20, Cd led to a decrease in PAO4 and PAO3 and an increase in PAO2 and PAO6, indicating different dynamics of individual PAOs in response to Cd exposure. This variable expression pattern either in individual PAOs at different time points or among various PAOs could be attributed to the develop-mental stages in the seedling, particularly due to the Cd stress condition. However, in PAO2 and PAO3, a similar trend may indicate that their specific role for Cd stress compares with the other PAOs. Nevertheless, conducting specific PAOs related mutants will provide clear insights into what extent different PAOs are involved in Cd stress resistance in the maize plant. The use of exogenous PAs or the use of inhibitors of enzymes involved in the PAs biosynthesis in traditional research techniques refers to the potential function of these substances in the defense and adaptability of plants to various environmental circumstances [16,57,58]. Several investigations have confirmed the protective effect of PAs in response to abiotic stress utilizing transgenic mutants in which their expression level is given, or their function is impaired [16,59,60]. High Put levels caused by homologous *arginine decarboxylase 1* (*ADC1*) overexpression improve *Arabidopsis*’ resistance to cold stress [61]. Similarly, the increase in Put induced by *arginine decarboxylase 2* (*ADC2*) overexpression induces drought tolerance in Arabidopsis, which could be related to a reduced water loss by stimulating stomatal closure [62]. In addition, the outcomes of lowering the expression of PA biosynthesis genes corroborate their protective function in plants in response to abiotic challenges. For instance, Arabidopsis spe1-1 and spe2-1 EMS mutants, which have decreased ADC activity, develop Put deficiency after being acclimated to high salt concentration and exhibit higher vulnerability to salt stress [63]. Arabidopsis acl5/spms double mutants, which do not produce Spm, are highly sensitive to salinity and drought stresses. The sensitivity of this phenotype is reduced by the application of exogenous Spm [64].

In plants, Cd stress exerts its effect through decreased miR390 levels as transgenic plants overexpressing miR390 by the constitutive 35S promoter are more sensitive to Cd shock than the wild-types, and transgenic plants also accumulated more Cd in roots [65,66,67,68]. The involvement of miR390 under Cd stress is verified in the current research. In contrast with miR390, plants overexpressing miR168a, the most commonly detected stress-inducible miRNA gene, display drought tolerance. Our data show that Cd exposure progressively reduces the miR390 level from the first date of application till day 12. Our results also indicate reduced Cd content in the root from day 12. Thus, the increase in miR390 from day 12 could be attributable to the Cd translocation from the root to the shoot. In this regard, on day 12, peroxisomal *PAOs* levels increased until day 12, followed by a reduction on day 15. In an opposite manner, miR390 expression and *PAOs* level can represent their negative feedback regulation under Cd stress. However, an increase in *POA1* content until day 20 can indicate the differential interaction of miR390 with apoplastic and peroxisomal *PAOs*. Our data show that in parallel with reduced miR390, Put levels ceased on day 12, followed by a remarkable increase on day 15. With regard to the negative role of miR390 in PAO enzyme, reduced Put level can be attributed to the PAOs elevation on day 12, which catalyzes Spd and Spm. This is consistent with enhanced Spd content, particularly on day 12. Accordingly, SPDS gene expression has been induced under Cd stress in a similar manner which implies a regulated controlling mechanism possibly associated with miR390. In parallel, reduced levels of Spm can be explained with the elevated *PAO3* expression level, which has been found to a play role in the back conversion reaction of Spm to Spd [53]. In higher plants, L-arginine is decarboxylated by ORDC and arginine decarboxylase (ARDC) to generate Put [69,70,71]. Our data showed that the *ORDC* expression was reduced by day 12 when the Put content was at the minimum level, while *ARGDC* represented a slight increase on the same date in plants supplemented with Cd. This suggests that the reduced level of Put on day 12 under Cd stress is most probably linked to the hampered expression of *ORDC*, which could be a result of the post-transcriptional regulatory mechanism of ORDC. However, the breakdown of ORDC is controlled by a protein called antizyme, which responds to the concentration of PAs. PAs decrease ORDC mRNA translation. Although ribosomal protein synthesis generally requires PAs, but in an opposite sense, the PAs excess also inhibits protein biosynthesis. Moreover, through an unraveled mechanism, ORDC is more sensitive to the PAs excess [72]. Whether this reduction can be associated with the reduced expression of miR390 is an interesting question, as in control plants *ORCD* was lower than contaminated condition [21,73].

The overexpression of the miR390′s close homolog, namely miR168, has shown drought stress sensitivity. miR168 homologs have been identified in various plant species, such as poplar (*Populus trichocarpa*), tobacco (*Nicotiana tabacum*), Arabidopsis, maize (*Zea mays*), and rice (*Oryza sativa*). Although miR168 homologs have been found to respond to various environmental stimuli, such as salt, drought, and cold stresses [74,75,76,77], the miR168 function under Cd stress has not been worked out yet. In the current study, Cd caused a reduction in the miR168 expression level, as the minimum content was found on day 12. However, a significant increase was observed afterwards. An interlink between an increase in the miR168 expression level and Cd exposure is unlikely as its expression level showed an elevation in the control condition as well. By contrast, miR528 was significantly up-regulated by Cd shock on day 6, however, its expression dropped to a minimum on day 12 and remained steady until day 20. The Cd responsive regulation of miR528 has been shown in rice (*Oryza sativa*). miR528 has been found to target the gene encoding *DCL1*, the RNA slicer enzyme, while miR168 was verified by its action to target a key component of the RISC complex in the miRNA pathway, namely *argonaute 1* (*AGO1*) [78]. Our results confirmed that miR528 expression is parallel with Put accumulation on day 6 by Cd application, indicating that, unlike miR390, Put induction is associated with miR528 up-regulation. This finding is consistent with the overexpression of *ORDC* on day 6. Besides, *PAOs*’ expression significantly is reduced on day 6. A plausible explanation could be that, on day 6, Cd accumulation in the root triggers a signaling cascade, including miR528 to suppress PAO functionality to prevent Spd and Spm biosynthesis. Interestingly, *PAO3*, mostly involved in PAs back conversion, has not been down-regulated by Cd shock, which implies a running back conversion reaction to produce Put, resulting in a higher Put level on day 6. In a different study, exogenous melatonin or caffeic acid O-methyltransferase (COMT) overexpression boosted the absorption and uptake of sulphur, which enhanced plant growth and Cd tolerance. This work presents genetic evidence that the effective control of S metabolism, redox homeostasis, and Cd translocation in tomato plants is intimately related to melatonin-mediated tolerance to Cd [79]. While several genes and metabolic pathways have been investigated so far, it is still unknown what their functions are and how they interlink to each other. The reciprocal synergy between functional genes should be extensively studied in future research, as the conventional method of studying a single gene can no longer satisfy the needs of the post-genomic age. Moreover, unidentified genes involved in plant Cd absorption and transport as well as their cooperative interactions, several genes’ roles and the connections between them, are still unclear. Furthermore, by creating mutants and utilizing molecular biology techniques in future studies, it will be possible to further investigate unidentified genes involved in plant Cd absorption and transport as well as the synergistic link between these genes.

Measurements of photosynthesis performance can be useful for assessing plant tolerance to stress [42]. In our study, the absorption rate in the reaction centers (ABS/RC) of PSII was inhibited by Cd exposure. This indicates that Cd disturbs the proportion of the absorbed light by chlorophyll, which is used for photochemistry. Excited electrons by light should be trapped by electron acceptors before turning down to the base status. The photosystem II (PSII) efficiency in the traping of the excited electron reflects the higher PSII capacity in transferring light to the electron transport chain [80,81]. Accordingly, the electron transport rate (ET0/RC) of PSII shows the oxidized quinone acceptor (Qa) as the energy consumer in the electron transport chain that protects photodamage [82]. If the electron transport chain cannot support the exited energy, energy dissipation (DI0/RC) reduces the electron shift from photochemically active centers to photochemically inactive PSII centers, which eventually will damage the light trapping process (TR0/RC), which will led to reduced PSII efficiency. This capacity protects photo-inactivated reaction centers from excessive light damage [83,84]. According to the present data, DI0/RC increased by Cd application in comparison to the control plants, which addresses the reduced protective potential by plants. Elevated DI0/RC demonstrates the harmful effect of Cd on photosynthesis functionality [21].

Higher NPQ in Cd-contaminated plants is indicative of the quenched singlet-excited chlorophylls (Chl) through the non-photochemical mechanism. A number of researchers have reported that reactive oxygen species are actively produced in photosystem I and photosystem II under unfavorable environmental conditions (PAs and abiotic stress tolerance in plants). The present findings revealed that Cd-reduced Fv/Fm is regarded as a crucial indicator of the plants’ photosynthetic status in stress conditions. In our study, Cd exposed plants showed lower Fv/Fm in comparison with the controls, indicating the negative role of the Cd on photosynthetic performance. The Pi-abs value, which reflects the functionality of both photosystems I and PSII, will also be diminished by the Cd application [85]. High NPQ represents a minimized O_2_ production in the PSII antenna [86] and in the current findings, Cd exposure results in lower NPQ and elevated O_2_ production. According to these results, Cd dynamically affects photosynthetic functionality by impairing the opening and closing of the reaction centers, electron transport, and energy consumption.

## 4. Conclusions

Cd has been found as a toxic heavy metal with adverse effects on plant growth and development. There is a wealth of information about the PAs roles in response when plants encounter Cd toxicity. However, knowledge about the dynamic role of PAs in plants in different developmental stages (time points) against Cd stress is still scarce. Current evidence suggests a role for PA and associated miRNAs. Although it is not yet clear why plants give priority to certain PAs in response to stress, our results show that at different time points the biosynthesis of various PAs is induced and this is associated with different miRNA related metabolic pathways under Cd contamination (Figure 8). This leaves one likely scenario, that Cd levels at different time points in the root, modulated by its uptake and transportation to the shoot, function as a signal to induce a dynamic PA/miRNA response in maize plants. For example, PA content variation showed that Cd accumulation led to an increase in PA, while Cd uptake on day 3 did not change significantly the Pas content.

Nevertheless, further studies such as determining the mechanism of Cd uptake and transport are warranted to elucidate a robust mechanism/s underlying the rise of Cd stress responses. This will pave the way for controlling Cd uptake to inhibit the detrimental effects of Cd on the different plant growth and developmental stages.

## 5. Materials and Methods

### 5.1. Plant Material and Growth Condition

Maize (*Zea mays* L. variety “260”) seeds were obtained from Karaj Agricultural Research Institute (Tehran, Iran). Healthy and uniform seeds were surface-sterilized with 1% (*v*/*v*) sodium hypochlorite solution and rinsed 3 times with sterile distilled water. Each sterilized seed was sown in a bucket of culture trays with a 7 cm depth containing 6 g of coco-peat. 3 g of coco-peat was added to each bucket to cover the seeds. The culture trays were placed under controlled conditions at 23 ± 2 °C, 65% relative humidity and 16/8 h light-dark photoperiod for 20 days. Daily irrigation was applied for each bucket with 3 mL of water. Each culture tray was placed in a plastic bag to prevent possible leakage. For Cd treatment, one kg coco-peat was mixed with 0 and 150 mg CdCl_2_. To balance the water in treated coco-peat, pots were kept for 48 h at 23 ± 2 °C. The seed germination was measured after 3 days of sowing and the uniform-sized seedlings were selected for further analysis on 6, 12, 15 and 20 days post-sowing. Each treatment and each time point (30 plants) was replicated three times. Two cm from the root tip were harvested and transferred to liquid nitrogen and stored at −80 °C for measurement of physiological and molecular parameters prior to analysis.

### 5.2. Determination of Plant Growth Parameters

At each time-point, root and shoot lengths and leaf area were measured by analyses of the scanned images using ImageJ software (version 1.44P; US National Institutes of Health, Bethesda, MA, USA) [87].

### 5.3. Cd Determination

Cd content was measured in both root and shoots at 3, 6, 12, 15 and 20 days after seed sowing. Plant tissue was dried at 70° C for 24 h and weighed. The dried tissue was incubated in an oven at 500–700 °C for 4 h. After cooling, ashes were dissolved in 5 mL nitric acid for 18 h [88]. The solution was filtered on a Whatman filter paper and washed with distilled water to obtain a final volume of 10 mL. Cd concentration was determined by atomic absorption/flame emission spectrophotometer (Shimadzu, Japan). Cd concentration (mg/L) of each sample was calculated using the standard curve prepared with a range of known concentrations. Cd content (mg/kg plant dry weight) was calculated by the following equation. The equipment was sourced from Shahid Beheshty University of Tehran, Iran.
Cd content (mg/kg) = [Cd concentration (mg/L) × final volume of sample (L)]/dry weight of tissue (kg)

Cd translocation factor was calculated by dividing the Cd concentration of the shoot by the Cd concentration of the root [89].

### 5.4. Transient and Slow Induction of Chlorophyll Fluorescence

Outermost leaves following 20 days of application of Cd were used for measuring slow chlorophyll fluorescence parameters. After dark adaptation for at least 20 min, intact attached leaves to the plants were immediately used to measure chlorophyll fluorescence parameters using a fluorometer system (Handy FluorCam FC 1000-H Photon Systems Instruments, PSI, Czech Republic). Images were recorded during short measuring flashes in darkness. At the end of the short flashes, the samples were exposed to a saturating light pulse (3900 µmol m^−2^ s^−1^) that resulted in a transitory saturation of photochemistry and reduction of the primary quinone acceptor of PSII [90,91,92]. After reaching steady-state fluorescence, two successive series of fluorescence data were digitized and averaged, one during short measuring flashes in darkness (F_0_), and the other during the saturating light flash (Fm). From these two images, variable fluorescence (Fv) was calculated according to the ratio between Fm and F_0_. The Fv/Fm was calculated using the ratio between F_v_ and F_m_. Maximum fluorescence in light-adapted steady state (Fm′) was determined and was used for the calculation of NPQ based on the following equation:NPQ = (Fm/Fm′) − 1

From these data, the following parameters were calculated: the specific energy fluxes per reaction centre (RC) for energy absorption (ABS/RC = M_0_·(1/V_J_)·(1/φ_Po_)), (M_0_ = TR_0_/RC − ET_0_/RC), trapped energy flux (TR_0_/RC = M_0_·(1/V_J_)), electron transport flux (ET_0_/RC = M_0_·(1/VJ)·ψ_o_) and dissipated energy flux (DI_0_/RC = (ABS/RC) − (TR_0_/RC)).

### 5.5. Measurement of H_2_O_2_ Content

H_2_O_2_ was determined by homogenizing plant fresh tissue (500 mg) in 5 mL trichloroacetic acid (TCA) 1% on ice and centrifuged at 12,000 × *g* for 15 min at 4 °C. 500 µL of the supernatant was mixed with 500 µL of 10 mM potassium phosphate buffer (pH 7) and 1 mL of 1 mM potassium iodide. The absorbance of the solution was measured spectrophotometrically at 390 nm [93].

### 5.6. PAs Determination

PAs were extracted according to the method of [94] with some modifications. Briefly, 0.2 g fresh weight was homogenized with 1 mL of 5% (*v*/*v*) perchloric acid (PCA) and was centrifuged at 15,000× *g* for 35 min at 4 °C. Then, the precipitate was used for the extraction of bound PAs and the supernatant was used for the extraction of conjugated and free PAs. For extraction of bound PAs, the precipitate was washed three times with 5% PCA to extract pure bound PAs. Then, 1 mL of 1N NaOH was added to the precipitate and centrifuged at 21,000× *g* for 30 min at 4 °C and 100 µL of the supernatant was hydrolyzed with 200 µL of 6N HCl and inserted into the ampoule. For extraction of conjugated PAs, 100 µL of the supernatant was hydrolyzed with 200 µL of 6N HCl and inserted into the ampoule. To evaporate the acid, the ampoules were incubated in an oven at 110 °C for 18 h. Then, the tops of the ampoules were broken, followed by incubation at 80 °C for 18 h. For dansylation, 100 µL of the hydrolyzed PCA supernatant containing conjugated PAs or 100 µL of the hydrolyzed pellet containing bound PAs were mixed with 200 µL of saturated sodium bicarbonate (Na_2_CO_3_) and dansyl chloride (5 mg·mL^−1^) and incubated at 60 °C for 90 min. Excess dansyl reagent was removed by reaction with 200 µL of added proline (0.1 g·mL^−1^) and incubation at 60 °C for 45 min. For dansylation of free PAs, the above procedure, except for the addition of Na_2_CO_3,_ was conducted with unhydrolyzed PCA supernatant. Dansyl-PAs were extracted in 500 µL toluene. The upper phase was collected, and the PAs were separated by thin layer chromatography, performed on high-resolution silica gel plates using an n-hexane: ethyl acetate (4:5) solvent system. Silica plates were observed under UV light. Dansylated PAs were identified by comparing the Rf values of dansylated standards.

### 5.7. Total RNA Extraction

Total RNA was isolated according to the method of Jazi et al. (2015) with slight modifications [95]. Briefly, 70 mg root samples were mixed with 1 mL of extraction buffer (2% CTAB, 100 mM Tris-HCl, 2 M NaCl, 25 mM EDTA, 2% PVP and 4% β-mercaptoethanol, pH 8) and homogenized by vortexing for 40 s and incubated at 65 °C for 35 min. Then, an equal volume of chloroform: isoamyl alcohol (1:24 *v*/*v*) solution was added to the homogenized solution and incubated for 15 min at room temperature. The upper phase was separated by centrifugation at 5000× *g* for 20 min at 4 °C and then mixed with an equal volume of 5 M NaCl and cold isopropanol and incubated at −20 °C for 18 h. Then, the solution was centrifuged at 20,000× *g* for 35 min at 4 °C. The supernatant was removed, and the pellet was suspended with 50 µL DEPC water and 113 µL of 12 M LiCl and incubated for 18 h at 4 °C. The RNA pellet was washed using 5 M NaCl and cold 70% ethanol and finally, the pellet dissolved in 40 µL of DEPC water. The quality and quantity of the extracted RNA were verified by 0.8% agarose gel electrophoresis and spectrophotometer, respectively. The extracted RNA was treated with RNase-free DNase I (Qiagen, Tehran Iran), to remove possible genomic DNA contamination.

### 5.8. miRNA Expression Analysis via Stem-Loop Reverse Transcription

The sequences of microRNAs were downloaded from miRBase database (version 22.0, http://www.mirbase.org, accessed on 28 March 2019) (Table 1). The miRBase accession number for the microRNAs is listed in Table 2. To assay miR390, miR168 and miR528 expression, the miRNAs were reverse transcribed into cDNAs by stem-loop reverse transcriptase primers (Table 2), which were designed following the methods described by Chen et al. (2005) and Liu et al. (2012) [96,97]. The last complimentary six nucleotides at the 3′ end of miRNA were added to the 3′ ends of the backbone of stem-loop RT primer. The backbone sequence of stem-loop RT primer for miR390, miR168 and miR528 are GTCGTATCCAGTGCAGGGTCCGAGGTATTCGCACTGGATACGAC [97] and for *U6* as internal control is GTCGTATCCAGTGCAGGGTCCGAGGTATTCGCACTGGAT ACGACAAAATA [98]. The RT reactions were performed using cDNA Synthesis Vivantis kit (2-step RT-PCR kit, Vivantis Technologies, Setia Alam, Malaysia) according to the manufacturer’s instructions. The qPCR of miR390, miR168 and miR528 was carried out on the miRNA-specific RT product as a template with a miRNA-specific forward primer and a universal reverse (complementary to the backbone sequence of the RT primer) primer. *U6*, one of the uniformly expressed small RNAs, was used as the internal control for stem-loop RT-PCR. The sequence of stem-loop RT and qPCR primers were listed in Table 2.

### 5.9. Gene Expression Analysis by RT-qPCR

The maize sequences of 6 *PA oxidases* (*PAOs*), *Arginine decarboxylase* (*ARGDC*), *Ornithine decarboxylase* (*ORDC*) and *Spermidine synthase* (*SPDS*) genes involved in PAs metabolism and a housekeeping gene i.e., *membrane protein PB1A10.07c* (*MEP*) as the internal control [99] were derived from NCBI (http://www.ncbi.nlm.nih.gov/ accessed on 28 June 2019) or phytozome (https://phytozome.jgi.doe.gov/ accessed on 28 June 2019). The accession number for the genes are listed in Table 3. To analyze the gene expression, the extracted RNA using the cDNA Synthesis Vivantis kit with the oligo-dT primer was reverse transcribed to cDNA according to the manufacturer’s instructions. All specific primers used for real-time qPCR were from the full-length cDNA sequences of the genes and are listed in Table 3.

### 5.10. Real-Time Quantitative PCR

All real-time qPCR was performed using SYBR Green PCR Master Mix (BIOFACT, Republic of Korea) on a Step One Plus™ Real-Time PCR System (Applied Biosystems, USA). The reactions were amplified for 15 min at 95 °C, followed by 40 cycles of 95 °C for 20 s, 59–60 °C for 30 s and 72 °C for 15 s. The PCR products of the miRNAs and genes were checked on 3% and 1% agarose gels, respectively. Reactions were performed on three biological (RNA) and three technical replicates. Normalized expression levels of the miRNAs and genes were calculated with 2^−ΔΔCt^ method using *U6* and *MEP* as the internal control, respectively. The data were analyzed according to [100].

### 5.11. Statistical Analysis

All experiments were performed in three biological replicates and a minimum of three technical replicates. The data were presented as the mean ± standard error (SEM). The differences in the mean values were statistically analyzed by PRISM software (version 7.01) using a two-way analysis of variance (ANOVA), followed by the Tukey test (*p* < 0.05).

## Figures and Tables

**Figure 1 plants-12-01991-f001:**
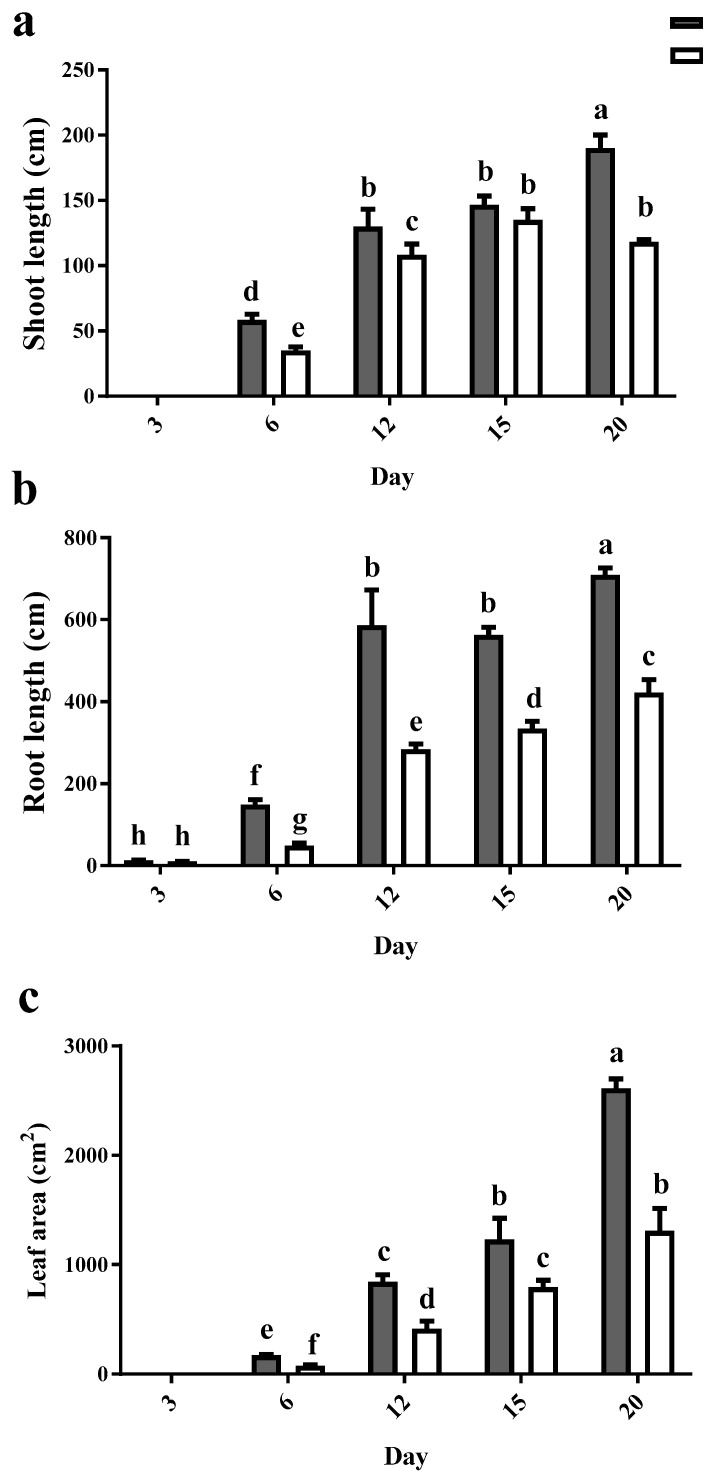
Effects of Cd [0 mg/kg (control) and 150 mg/kg CdCl_2_] on shoot length (**a**), root length (**b**) and leaf area (**c**) of maize plants over 20 days after CdCl_2_ exposure. Each column represents the mean value of three replications ± standard error. Bars with different letters are significantly different (ANOVA, *p* < 0.05).

**Figure 2 plants-12-01991-f002:**
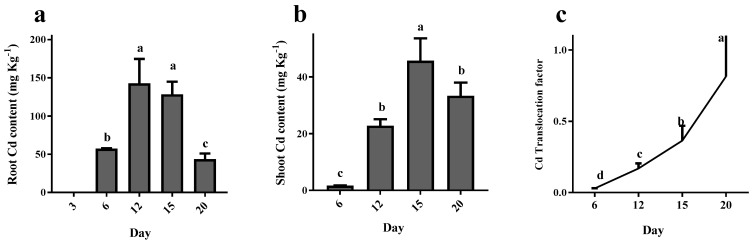
Cd concentration in root (**a**), shoot (**b**) and Cd translocation factor (**c**) during 20 days of maize plant exposure to 150 mg/kg CdCl_2_. Each column represents the mean value of three replications ± standard error obtained from at least three technical replicates. Bars with different letters are significantly different (ANOVA, *p* < 0.05).

**Figure 3 plants-12-01991-f003:**
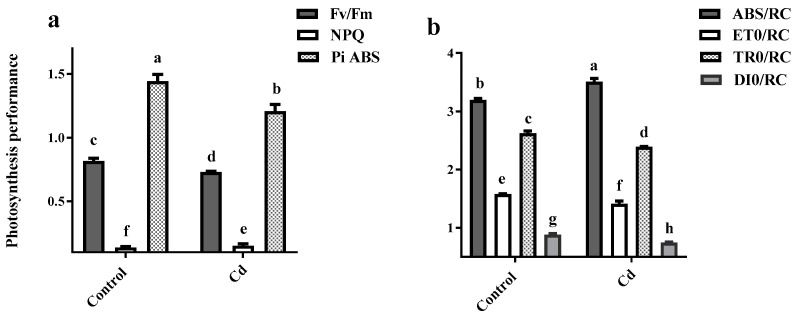
Photosynthetic performance (**a**) [maximum quantum efficiency of photosystem II (Fv/Fm), non-photochemical quenching (NPQ) and performance index on absorption basis (Pi-ABS)] and energy fluxes [per reaction center (RC)]. (**b**) for energy absorption (ABS/RC), electron transport (ET_0_/RC), trapped energy (TR0/RC) and dissipated energy (DI_0_/RC) in maize plants following 20 days exposed to 0 mg/kg (control) and 150 mg/kg CdCl_2_. Each column represents the mean value of three replications ± standard error obtained from at least three technical replicates. Bars with different letters are significantly different (ANOVA, *p* < 0.05).

**Figure 4 plants-12-01991-f004:**
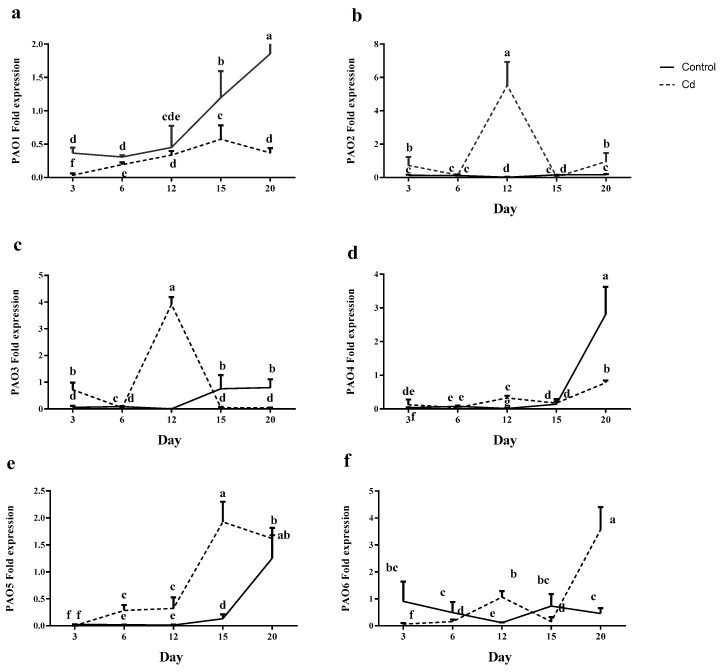
Relative expression of (**a**) apoplastic *polyamine oxidase* (*PAO*) *1*, (**b**) peroxisomal *PAO2*, (**c**) peroxisomal *PAO3*, (**d**) peroxisomal *PAO4*, (**e**) apoplastic *PAO5* and (**f**) peroxisomal *PAO6* in the root of maize plants after 3, 6, 12, 15 and 20 days exposed to 0 mg/kg (control) and 150 mg/kg CdCl_2_. Each column represents the mean value of three replications ± standard error obtained from at least three biological and three technical replicates. Bars with different letters are significantly different (ANOVA, *p* < 0.05).

**Figure 5 plants-12-01991-f005:**
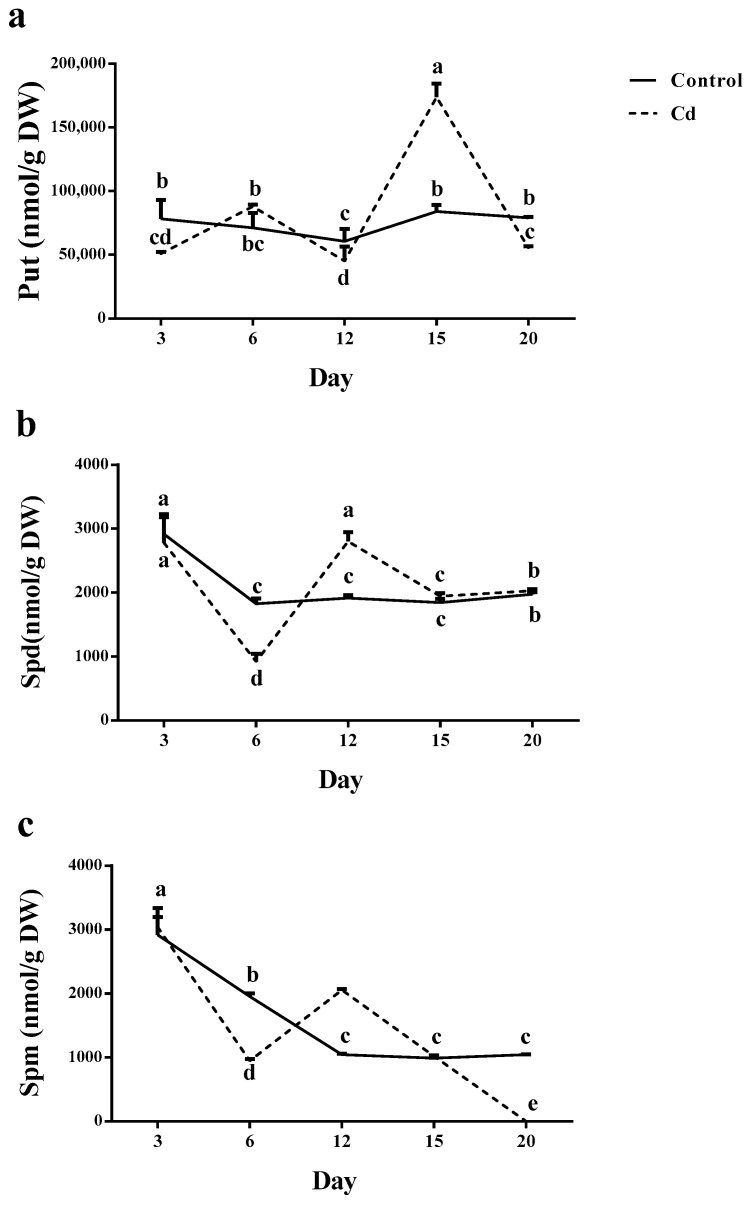
Effect of cadmium (Cd) application [0 mg/kg (control) and 150 mg/kg CdCl_2_] on a different form of polyamine content [putrescine (Put, (**a**)), spermidine (Spd, (**b**)) and spermine (Spm, (**c**))] in the root of maize plants after 3, 6, 12, 15 and 20 days exposed to 0 mg/kg (control) and 150 mg/kg CdCl_2_. Each column represents the mean value of three replications ± standard error obtained from at least three technical replicates; each value represents mean ± standard error obtained from three technical replicates. Bars with different letters are significantly different (ANOVA, *p* < 0.05).

**Figure 6 plants-12-01991-f006:**
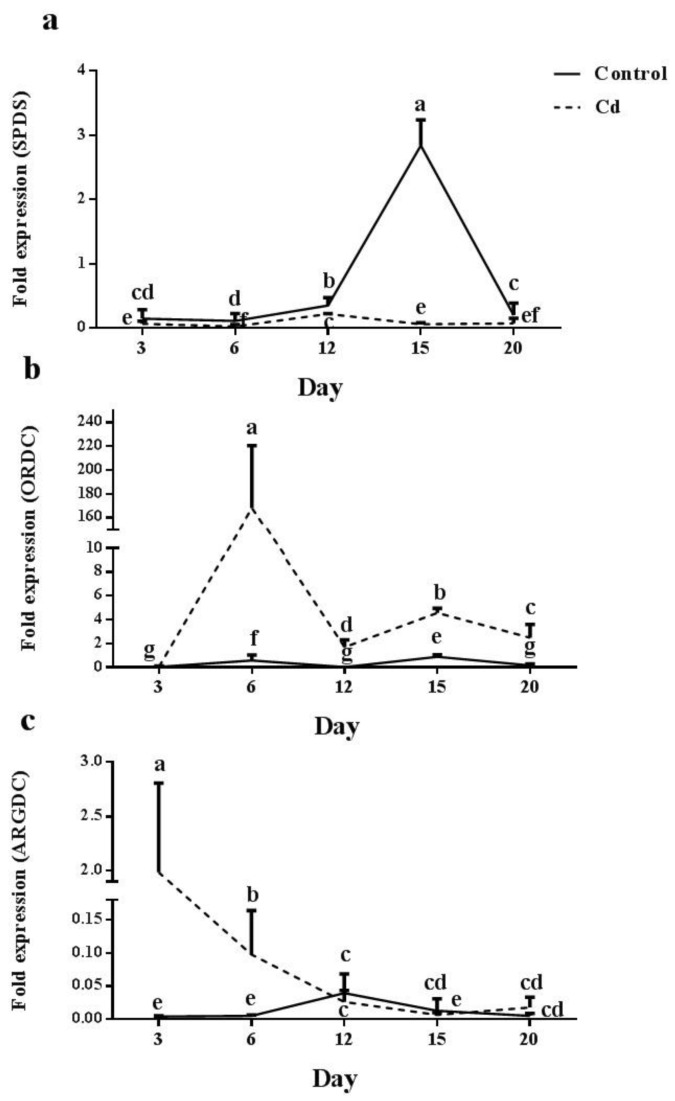
Relative expression levels of *spermidine synthase* (*SPDS*) (**a**), *ornithine decarboxylase* (*ORDC*) (**b**) and *arginine decarboxylase* (*ARGDC*) (**c**) genes in the root of in the root of maize plants after 3, 6, 12, 15 and 20 days exposed to 0 mg/kg (control) and 150 mg/kg CdCl2. Each column represents the mean value of three replications ± standard error obtained from at least three biological and three technical replicates. Bars with different letters are significantly different (ANOVA, *p* < 0.05).

**Figure 7 plants-12-01991-f007:**
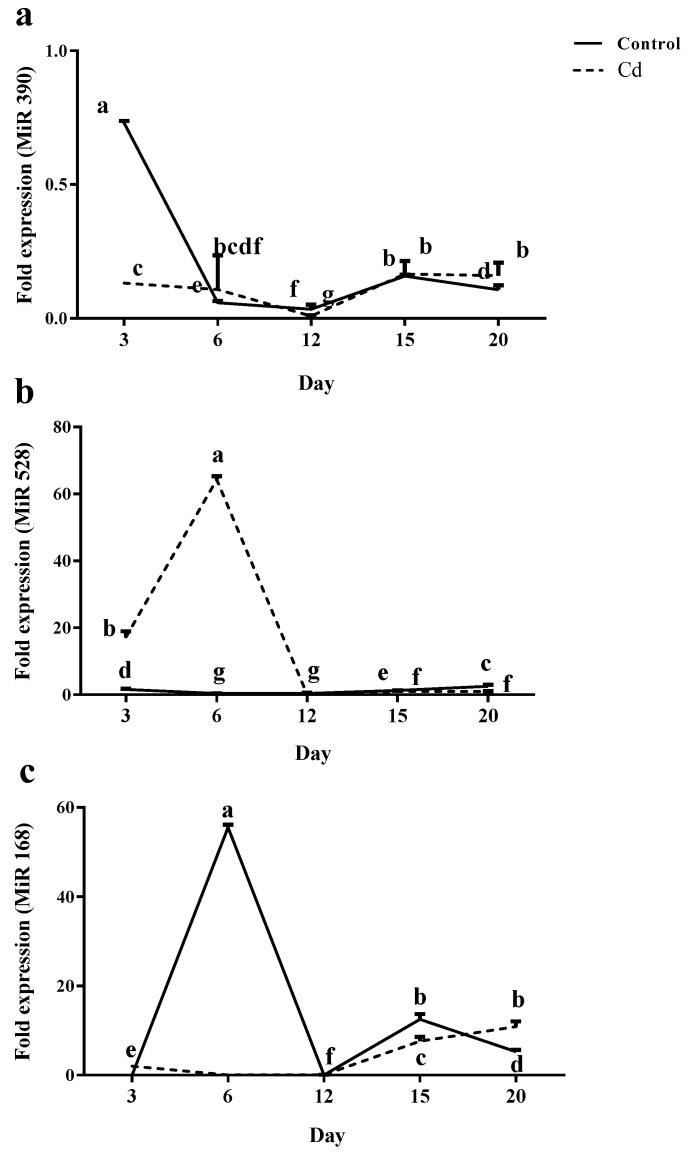
Relative expression levels of miRNAs [miR390 (**a**), miR528 (**b**) and miR168 (**c**)] in the root of maize plants after 3, 6, 12, 15 and 20 days exposed to 0 mg/kg (control) and 150 mg/kg CdCl_2_. Each point represents the mean value ± standard error obtained from three biological replicates and at least three technical replicates. Bars with different letters are significantly different (ANOVA, *p* < 0.05).

**Figure 8 plants-12-01991-f008:**
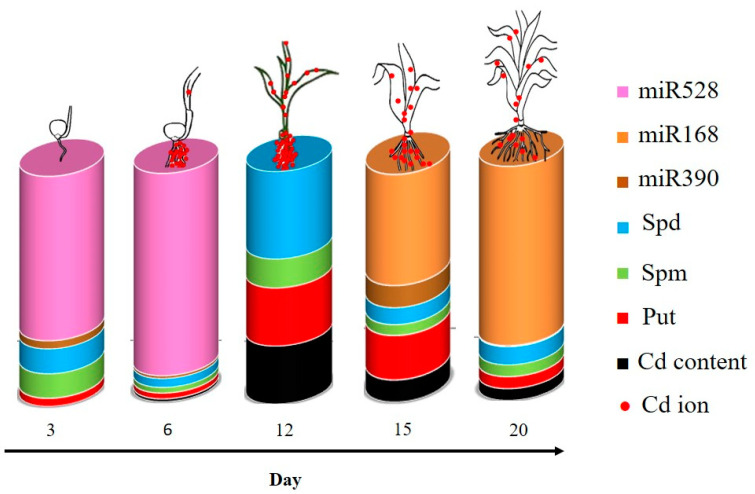
Schematic representation of PAs accumulation and miRNAs genes expression in response to the cadmium (Cd) stress in the root of maize plants after 3, 6, 12, 15 and 20 days exposed to 0 mg/kg (control) and 150 mg/kg CdCl_2_. According to the scheme, Cd accumulation hampers the primary elevated level of miR528 on day 12 and in opposite scenes, induced miR168 and miR390 levels, inhibiting Cd intake on days 15 and 20. However concurrent with the first and most detectable Cd accumulation on day 12, PAs levels increase significantly which can be linked to the further miR390 and miR168 expression. Red dots represent the Cd ion in the shoot and roots of maize at different time points. The scheme highlights the possibility that Cd levels at various time points in the root would be influenced by its uptake and transportation to the shoot, serve as a signal to trigger a dynamic PA/miRNA reaction in maize plants.

**Table 1 plants-12-01991-t001:** The sequence of mature miR390, miR168 and miR528.

miRNA	Sequence (5′-3′)
Mir390	AAGCUCAGGAGGGAUAGCGCC
Mir168	UCGCUUGGUGCAGAUCGGGAC
Mir528	UGGAAGGGGCAUGCAGAGGAG

**Table 2 plants-12-01991-t002:** List of stem-loop RT and qPCR primers for miRNA expression analysis.

miRNA (miRBase Accession Number)	Stem-Loop RT Primer Sequence (5′-3′)	qPCR Primer Sequence (5′-3′)
MiR390 (MIMAT0014033)	GTCGTATCCAGTGCAGGGTCCGAGGTATTCGCACTGGATACGACGGCGCT	Forward: CGACTGAAGCTCAGGAGGGAT
		Universal Reverse: GTGCAGGGTCCGAGGT
MiR168 (MIMAT0001726)	GTCGTATCCAGTGCAGGGTCCGAGGTATTCGCACTGGATACGACGTCCCG	Forward: CGACTGTCGCTTGGTGCAGAT
		Universal reverse: GTGCAGGGTCCGAGGT
MiR528 (MIMAT0014029)	GTCGTATCCAGTGCAGGGTCCGAGGTATTCGCACTGGATACGACCTCCTC	Forward: CGACTGTGGAAGGGGCATGCA
		Universal reverse: GTGCAGGGTCCGAGGT
*U6*	GTCGTATCCAGTGCAGGGTCCGAGGTATTCGCACTGGATACGACAAAATATGGAAC	Forward: TGCGGGTGCTCGCTTCGGCAGC
		Reverse: GGGCAGCCAAGGATGACT

**Table 3 plants-12-01991-t003:** List of qPCR primers for gene expression analysis.

Gene	Accession No. (NCBI Gene Bank or Phytozome)	Forward Primer Sequence (5′-3′)	Reverse Primer Sequence (5′-3′)
ARGDC	GRMZM2G100920 (NM_001365614.1)	CTAATATGCCCGTATCCACC	GGCAATCATCATAAGTCGCAC
ORDC	NM_001148682.1	CATGGACCACAAGGCTCC	GTCGAAGACGAGCCAGTC
SPDS	AY730048.1	CGAAAGAATCAGTGTCAGAACC	GTGCGGTGTCAGCAAAAGC
PAO1	GRMZM2G034152 (NM_001111636.2)	GCAAGTACCATGTCCAGGG	CGAGGGAACATGGCTGTCA
PAO2	GRMZM2G000052_T01	TGGAGATGTGCCACCCTG	GGCAATCAGTGGGATGTCC
PAO3	GRMZM2G396856_T02	GACGAAAGCCCTGTCTCC	CGAAGAGGGAGAAGCAAGG
PAO4	GRMZM2G150248_T01	TCCTACTCGTGCGACCTG	CGATGCCTGACGAGTAAGC
PAO5	GRMZM2G035994_T01	CAGCACTACGCTTAGGTTGC	TAGCACACAGCAAGAACACAG
PAO6	GRMZM2G078033_T01	GATTTGCAGGACCTGAGTGAG	CAAGACACAACGGCCTTCAAG
MEP	GRMZM2G018103 T01	TGTACTCGGCAATGCTCTTG	TTTGATGCTCCAGGCTTACC

## Data Availability

The data presented in this study are available on request from the corresponding author. The data are not public.

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
