# Peer review of "Response to Cadmium Toxicity: Orchestration of Polyamines and microRNAs in Maize Plant"

_plants, 2023, doi:10.3390/plants12101991_

Round 1
Reviewer 1 Report
Comments to authors
The manuscript "Response to Cd toxicity: Orchestration of polyamines and microRNAs in maize plant" is interesting and innovative as it deals with some interesting mechanisms. The work done by the author are quite promising and can be a good addition to the scientific literature. I have the following minor comments which should be covered by the authors before its publication:
1) What are the reasoning for studying those specific polyamines, that information should be added to the introduction
2) Figure legends should be self-explanatory with enough information and the short title should be avoided. There would be information for the reader to provide the required information without referring to the text
3) The main message of the scheme should be also added to its figure legends
4) On the X axis's title of the figure's "day" is written. It should be clear what this exactly means. It should be clear it is days after cadmium exposure? or days after culturing of plants in general.
5) Lines 418-438: the information about the equations extracted from the OJIP protocol is already indicated in a previous publication, better to have a related reference and remove that part.
6) The importance of the mechanism underlying cadmium translocation to the shoot needs to be more elaborated in the discussion, does sequestration of cadmium by the root is beneficial for plant tolerance, elaborate this issue in the discussion.
7). The authors should add some more recent references related to the topic. Few examples are given below: the authors should add them in revision.
· DOI: 10.3390/antiox8120641
· doi: 10.3390/antiox11020225
doi: 10.1021/acs.jafc.9b02404.
Author Response
The manuscript "Response to Cd toxicity: Orchestration of polyamines and microRNAs in maize plant" is interesting and innovative as it deals with some interesting mechanisms. The work done by the author are quite promising and can be a good addition to the scientific literature. I have the following minor comments which should be covered by the authors before its publication:
- What are the reasoning for studying those specific polyamines, that information should be added to the introduction.
Author response: Thank you for your comments. We have provided the reasoning by adding a new paragraph in the introduction (lines 73-95).
- Figure legends should be self-explanatory with enough information and the short title should be avoided. There would be information for the reader to provide the required information without referring to the text
Author response: Agree. All figure legends were revised and supplemented with enough information.
- The main message of the scheme should be also added to its figure legends
Author response: Done.
- On the X axis's title of the figure's "day" is written. It should be clear what this exactly means. It should be clear it is days after cadmium exposure? or days after culturing of plants in general.
Author response: To make the message clear in the figures ''days after cadmium exposure'' was included to all legends.
- Lines 418-438: the information about the equations extracted from the OJIP protocol is already indicated in a previous publication, better to have a related reference and remove that part.
Author response: OJIP repeated protocol was removed according to the reviewer's suggestion and only the reference remained.
- The importance of the mechanism underlying cadmium translocation to the shoot needs to be more elaborated in the discussion, does sequestration of cadmium by the root is beneficial for plant tolerance, elaborate this issue in the discussion.
Author response: Thank you for your comments, a new section has been added to the discussion (Lines 279-291).
- The authors should add some more recent references related to the topic. Few examples are given below: the authors should add them in revision.
- DOI: 10.3390/antiox8120641
- DOI: 10.3390/antiox11020225
DOI: 10.1021/acs.jafc.9b02404.
Author response: Suggested references have been cited in related parts in the body of the manuscript.
Reviewer 2 Report
This manuscript explored the cellular mechanisms on the development of Cd resistance in maize. Research outcomes are informative, but the entire quality can be further improved in the following aspects.
1. Add a legend to indicate Control and Cd in Figure 4.
2. Include more discussion on the correlations among plant morphological characters, Cd accumulation, enzymes, and gene expressions. The changing trends during the experimental period and proper discussions are particularly of interest.
3. Explain the drop of Cd in both root and shoot from days 15 to 20. How about the biomass for each plant? Will the increased biomass (need data support) be a suitable explanation? Please include the data and work on the calculations to offer proper discussion.
Author Response
This manuscript explored the cellular mechanisms on the development of Cd resistance in maize. Research outcomes are informative, but the entire quality can be further improved in the following aspects.
- Add a legend to indicate Control and Cd in Figure 4.
Author response: All figure legends annotation statements were revised according to the reviewer's suggestion.
- Include more discussion on the correlations among plant morphological characters, Cd accumulation, enzymes, and gene expressions. The changing trends during the experimental period and proper discussions are particularly of interest.
Author response: Detailed discussion was included in the body of discussion ranging from Cd accumulation, sequestration, gene expression and Cd-PAs interaction.
- Explain the drop of Cd in both root and shoot from days 15 to 20. How about the biomass for each plant? Will the increased biomass (need data support) be a suitable explanation? Please include the data and work on the calculations to offer proper discussion.
Author response: The reduction of cadmium on the 20th day compared to the 15th day is due to the growth of the plant, which is well indicated in the data provided in Figure 1. The figure shows that in comparison to the 15th day on the 20th day, the root length and the leaf area increased significantly.
Reviewer 3 Report
The manuscript entitled “Response to Cd toxicity: Orchestration of polyamines and microRNAs in maize plant” investigated the mechanisms utilized by plants to cope with Cd toxicity. And the results confirmed that the plant growth and photosynthetic mechanism were negatively influenced during 20 days of Cd stress, and on day 12, the day that the plant launches Cd transport to the shoot, all miRNA expression is reduced. This topic has been attracted large amount of attentions, due to suggesting a plant-derived strategy to commit a polyamines /miRNA-regulated mechanism/s in different developmental stages (time points) in response to Cd exposure. The experiment is generally well designed, and the conclusion has been evidenced and supported by the data. However, the organization and writing should be carefully improved, please also go through the text. Some more revisions are needed before publication as following:
Major comments:
Please check the manuscript carefully, including spell and grammar. Choose the precise words for writing. And pay attention to the logical problems of writing.
The diagrams should be described and explained in more detail, for example, Figure 4 should add a legend for reading.
Use richer forms of vocabulary expression to improve the readability of your articles.
The introduction should be reorganized for a better structure.
Minor comments:
L52-54 Please divide this sentence to two.
L58 “favours” should be replaced by “favors”
L60“unfavourable”is better to be changed to “unfavorable”
L69“defence” should read “defense”
L75“as stress-related metabolites in plant remains under debate” should be replaced by “as plant stress-related metabolites, remain under debate”
L86“MiRNAs inhibit gene” can be changed to “The miRNAs inhibit gene”
L108“in the near future” is better to be changed to “shortly”
L122 “represent” should read “represents”
L131 remove “a”
L168“afterwards” should be replaced by “afterward”
L187 “in” should read “on”
L192 “spermidine (Spm, b)” is not match the figure
L200 “day” should read “days”
L216“miR390” should be replaced by “The miR390”
L221 “in” should read “at”
L244 “are” is better to be changed to “have”
L306“signalling” should be replaced by “signaling”
L317 remove “of”
L334 wrong comma usage
L378“by” is better to be changed to “with”
L379“on” should read “in”
L482“4 %” should remove spaces
L504 “is” should read “are”
L545 “of” can be changed to “in”
Author Response
The manuscript entitled “Response to Cd toxicity: Orchestration of polyamines and microRNAs in maize plant” investigated the mechanisms utilized by plants to cope with Cd toxicity. And the results confirmed that the plant growth and photosynthetic mechanism were negatively influenced during 20 days of Cd stress, and on day 12, the day that the plant launches Cd transport to the shoot, all miRNA expression is reduced. This topic has been attracted large amount of attentions, due to suggesting a plant-derived strategy to commit a polyamines /miRNA-regulated mechanism/s in different developmental stages (time points) in response to Cd exposure. The experiment is generally well designed, and the conclusion has been evidenced and supported by the data. However, the organization and writing should be carefully improved, please also go through the text. Some more revisions are needed before publication as following:
Major comments:
- Please check the manuscript carefully, including spell and grammar. Choose the precise words for writing. And pay attention to the logical problems of writing.
Author response: Manuscript is revised by a native English speaker.
- The diagrams should be described and explained in more detail, for example, Figure 4 should add a legend for reading.
Author response: Done
- Use richer forms of vocabulary expression to improve the readability of your articles.
Author response: We did our best to improve the writing quality of the manuscript
- The introduction should be reorganized for a better structure.
Author response: Introduction part was revised carefully according to the reviewer suggestion.
Minor comments:
Author response: all minor comments have been corrected accordingly.
- L52-54 Please divide this sentence to two.
Author response: Suggested sentence was changed according to the manuscript revision by a native English speaker.
- L58 “favours” should be replaced by “favors”
Author response: ''indirectly favours ROS'' changed to ''indirectly favors ROS''
- L60“unfavourable”is better to be changed to “unfavorable”
Author response: ''unfavourable conditions'' changed to ''unfavorable conditions''
- L69“defence” should read “defense”
Author response: Sentence including ''defence'' was removed according to introduction re-statement.
- L75“as stress-related metabolites in plant remains under debate” should be replaced by “as plant stress-related metabolites, remain under debate”
Author response: Suggested sentence by the reviewer was included in the manuscript.
- L86“MiRNAs inhibit gene” can be changed to “The miRNAs inhibit gene”
Author response: ''The miRNAs inhibit gene'' changed to “MiRNAs inhibit gene”
- L108“in the near future” is better to be changed to “shortly”
Author response: “in the near future” changed to “shortly”
- L122 “represent” should read “represents”
Author response: “represent” changed to “represents”
- L131 remove “a”
Author response: “a” was removed from the sentence
- L168“afterwards” should be replaced by “afterward”
Author response: “afterward” changed to “afterwards”
- L187 “in” should read “on”
Author response: “in” changed to “on”
- L192 “spermidine (Spm, b)” is not match the figure
Author response: Spm changed to Spd in Fig. 5 legend
- L200 “day” should read “days”
Author response: ''day'' changed to ''days''
- L216“miR390” should be replaced by “The miR390”
Author response: “miR390” changed to “The miR390”
- L221 “in” should read “at”
Author response: ''In control'' changed to ''at control''
- L244 “are” is better to be changed to “have”
Author response: ''PAOs are localized'' changed to ''PAOs have localized''
- L306“signalling” should be replaced by “signaling”
Author response: ''a signalling cascade'' changed to ''a signaling cascade''
- L317 remove “of”
Author response: “of” was removed from the sentence
- L334 wrong comma usage
Author response: Comma was removed from the sentence
- L378“by” is better to be changed to “with”
Author response: “by” changed to “with”
- L379“on” should read “in”
Author response: “on” changed to “in”
- L482“4 %” should remove spaces
Author response: Space removed from “4 %”
- L504 “is” should read “are”
Author response: “is” changed to “are”
- L545 “of” can be changed to “in”
Author response: “of” changed to “in”
Reviewer 4 Report
I have reviewed the following manuscript entitled “Response to Cd toxicity: Orchestration of polyamines and microRNAs in maize plant” which was submitted in plants; my suggestions and comments are as follow:
“an experiment was conducted on maize seedlings.” Use the scientific name of the plant which have been studied in your experiment.
Keywords should not be repeated which already mentioned in the title.
“Cadmium (Cd) is one of the most persistent trace metals and a hazardous environmental pollutant that causes diverse biochemical and physiological malfunctions in plants, animals and humans” just focus on the plants not human and animal.
“The long half-life of Cd (10 to 30 years) raises concern about its accumulation in the body and its risk to human health” same comment.
“followed by an 18% decrease” grammar error.
“However, in the present work when plants were exposed to Cd contamination, shoot length increased until day 15 followed by a significant cessation by day 20.” How shoot length increased the Cd stress?
“Our previous study showed that maize plants exert mechanisms underlying PAs-related responses to cope with Cd shock” then what’s the novelty in the present study?
“Maize (Zea mays L. variety ''260'') seeds were obtained from Karaj Agricultural Research Institute (Tehran, Iran).” Is there any previous study on this variety?
“Plant tissue was dried at 70°C for 24 h and weighted. The dried tissue was incubated in an oven at 500-700°C for 4 h.” Add space between Celsius and degree.
Where is the conclusion section?
Author Response
Comments and Suggestions for Authors
I have reviewed the following manuscript entitled “Response to Cd toxicity: Orchestration of polyamines and microRNAs in maize plant” which was submitted in plants; my suggestions and comments are as follow:
“an experiment was conducted on maize seedlings.” Use the scientific name of the plant which have been studied in your experiment.
- Keywords should not be repeated which already mentioned in the title.
Author response: keywords were changed to Cadmium stress; MicroRNAs; Photosynthesis; Putrescine; Zea mays L according to the reviewer's suggestion.
- “Cadmium (Cd) is one of the most persistent trace metals and a hazardous environmental pollutant that causes diverse biochemical and physiological malfunctions in plants, animals and humans” just focus on the plants not human and animal.
Author response: ''animals and humans'' was removed from certain sentences.
- “The long half-life of Cd (10 to 30 years) raises concern about its accumulation in the body and its risk to human health” same comment.
Author response: The mentioned sentence was removed from the text.
- “followed by an 18% decrease” grammar error.
Author response: ''followed by an 18% decrease'' changed to ''followed by 18% decrease''
- “However, in the present work when plants were exposed to Cd contamination, shoot length increased until day 15 followed by a significant cessation by day 20.” How shoot length increased the Cd stress?
Author response: Based on the stem length results in Fig1, the negative effect of cadmium on stem growth was observed at each time point, although the plant under cadmium stress until day 15 showed an increase in growth compared to the previous time points, and then significant growth of the plant under cadmium stress did not show on day 20 compared to day 15. It seems that on the 20th day, a strong increase in the translocation of cadmium led to the cessation of plant growth compared to the 15th day. We have added this information in the discussion part that “Cd sequestration strategy in plants can harbor (perhaps partial in this case) resistance in plants”. Although revolutionary events can manipulate this tolerance to commit with Cd differentially. In our case, 20 days seems a biological threshold, nevertheless more in-depth experiments are required to fulfil this hypothesis.
- “Our previous study showed that maize plants exert mechanisms underlying PAs-related responses to cope with Cd shock” then what’s the novelty in the present study?
Author response: In the previous study we investigate the plants' response to Cd toxicity in PAs context. We have developed the current experiment based on the raised question that ''how'' PAs can induce plant tolerance. Therefore, in this study, we performed different time points (3, 6, 12, 15 and 20 days after Cd exposure) and MicroRNAs (Mir168, Mir390 and Mir528) gene expression analysis to get an insight into PAs-related plant-Cd-tolerance mechanism.
- “Maize (Zea mays L. variety ''260'') seeds were obtained from Karaj Agricultural Research Institute (Tehran, Iran).” Is there any previous study on this variety?
Author response: Below are the letters regarding previous studies on variety ''260''
- Among the three corn genotypes, the single cross (SC) 260 genotype is better than the SC600 and SC302 genotypes under the environmental condition. (Rabbani, B., & Safdary, A. (2021). Effect of sowing date and plant density on yield and yield components of three maize (Zea mays L.) genotypes in Takhar climatic conditions of Afghanistan. Asian J. Plant Sci. Innov, 1(2), 109-120.)
- Among eight varieties of maize (single cross 260, 301, 302, 500, 604, and 647 and double cross 370), roots of variety 260 had the greatest concentration of Pb as heavy metal (Tafvizi, M., & Motesharezadeh, B. (2014). Effects of lead on iron, manganese, and zinc concentrations in different varieties of maize (Zea mays). Communications in soil science and plant analysis, 45(14), 1853-1865). Since this information does not add value to the manuscript, they were not added to the revised manuscript
- “Plant tissue was dried at 70°C for 24 h and weighted. The dried tissue was incubated in an oven at 500-700°C for 4 h.” Add space between Celsius and degree.
Author response: Space was added between Celsius and degree
- Where is the conclusion section?
Author response: The conclusion section was added to the body of the manuscript.
Reviewer 5 Report
The authors have done a lot of experimental work and obtained interesting data. However, the conclusions made by the authors, even in the descriptive part, do not correspond well to the data presented in the graphs.
abstract
"Peroxisomal Polyamine oxidases (PAOs) PAO2, PAO3, PAO4, and PAO6 expression levels were increased in 12 days following Cd exposure".
This statement is inaccurate. A strong increase only for PAO2 and PAO3, and a transit one. In PAO4 and PAO6, it is almost imperceptible, if compared with their increase at later dates, if at all. That is, the dynamics of these 2 groups is completely different.
"Cd toxicity in days 6 and 15 caused an increase in Ornithine decarboxylase expression, which is involved in putrescine biosynthesis, leading to an elevation in putrescine levels in days 6 and 15."
Ornithine decarboxylase transiently increased on day 6 and putrescine only on day 15, which makes the relationship somewhat problematic as both were low on day 12 (at least this delay needs to be explained). In addition, the maximum Cd in the roots is just on the 12th day, and in the shoots on the 15th.
"Spermidine and Spermine levels were reduced on day 6 by Cd application which was parallel with suppressed Spermidine synthase gene. However, an increase in Spermidine and Spermine levels was observed on day 12 along with a significant elevation in Spermidine synthase expression."
According to Fig. 5cd, Spermidine and Spermine levels did decrease for a short time on day 6, but already on day 12 this decrease was more than compensated. According to Fig. 6a, no significant decrease in Spermidine synthase gene at day 6 is imperceptible, as well as a significant increase at day 12. The level of the enzyme during the treatment with Cd was slightly reduced throughout the experiment, incl. on 12 days. The most striking detail of the dynamics is the prevention of an increase in the enzyme in the control at day 15, which has little to do with Spermidine and Spermine levels.
"On day 6, Cd was observed to start being accumulated in the root and increased expression of microRNA 528; while on day 15, Cd started to be observed in the shooting part which an increase in microRNA 390 and microRNA 168."
According to Fig. 2ab, on day 6, Cd began to accumulate in the roots, but in shoots, the “beginning” of accumulation should rather be considered as 12 days, and 15 days is the maximum content of the toxicant. Transient expression of microRNA 528 in experimental plants really corresponds to 6 days, but on day 15, no accumulation of microRNA 390 and microRNA 168 occurred compared to the control.
"On day 12, the day that the plant launches Cd transport to the shoot, all miRNA expression is reduced."
On day 12, the minimum value was observed in the control. There was no decrease in microRNA 528 and microRNA 168 in the experiment compared to the control at all, and for microRNA 390 a slight decrease is most likely just an error in the use of statistics.
Discussion
L257-260 Greater cadmium accumulation does not imply greater cadmium sensitivity, and the relationship between cadmium accumulation and high miR390 content does not imply that the mechanism of cadmium toxicity is due to a decrease in miR390 content.
L263-266 " Our data represent that Cd exposure progressively reduces the miR390 level from the first date of application till date 12. Our results also indicate reduced Cd content in root from day 12. Thus, the increase in miR390 from day 12 could be attributable to the Cd translocation from the root to the shoot."
It follows from Fig. 7a that the level of miR390 is very high in young control plants and drops sharply with age. At the same time, the Cd curve almost completely coincides with the control, except for the starting point (3 days): At points 6, 12 and 20, the level in the Cd variant only slightly fluctuates around the control curve. Since the statistical analysis carried out by the authors confirms significant differences at a number of points, the question arises of how the biological replicates were obtained? Is biological repetition really biological. This can happen if samples for different replications were collected not from the material of individual plants, but from the total sample. If this is the case, then the total variance is underestimated and insignificant differences become significant. When interpreting experimental data, such a danger should be taken into account.
It seems to me that the following interpretation of the data in Fig. 7a is the most probable. 1) In control, miR390 levels are high in young seedlings and then drop sharply, 2) Treatment dramatically reduces miR390 levels in young seedlings within 3 days post-treatment (miR390 functions are associated with the early period and this is where Cd toxicity appears - however data is limited to one point and require confirmation), 3) after 12 days, miR390 increases slightly regardless of the presence of Cd and its transport.
L266-267 "In this regard, on day 12, peroxisomal PAOs levels increased until day 12 followed by a reduction on day 15".
From fig. 4bc, it can be concluded that this is a transitive increase in 2 out of 4 peroxisomal PAOs.
L267-268 "In an opposite manner, miR390 expression and PAOs level can represent their negative feed-back regulation under Cd stress."
The decrease in miR390 on day 12 occurs independently of Cd, the decrease lasts for several days, in contrast to the transitive increase under the action of Cd in 2 peroxisomal PAOs, i.e., most likely, they are not connected in any way.
L292-296 "In the current study, Cd caused a reduction in miR168 expression level, as the minimum content was found on day 12, however, a significant increase was observed afterwards. Interlink between an increase in miR168 expression level and Cd exposure is unlikely as its expression level showed an elevation in the control condition as well."
Judging by Fig. 7c, the most interesting moment in the behavior of miR168 is the transit peak on day 6 in the control, i.e. a little later than miR390. However, in both cases, their transit increase is almost completely suppressed by treatment with Cd. So, here the relationship with stress is obvious.
Minor comments
L54-56 "In addition, Cd can be easily replaced by similar divalent cations in the active sites of antioxidant proteins, leading to 55 an imbalance in cellular reactive oxygen species (ROS)". Please fix the passive form (can easily replace).
L58-59 "The long half-life of Cd (10 to 30 years) raises concern about its accumulation in the body and its risk to human health [5]".
Specify what half-life you are talking about: Cd is not a radioactive element.
Methods
How many plants are in each replication?
4.3. Cd determination Indicate the weight of the sample and how many plants the tissue for analysis was taken from.
Fig.2 What is the reason for the simultaneous decrease in the concentration of cadmium in the roots and shoots on the 20th day?
Fig. 3. Specify the measurement time. In the methodology: "4, 8 and 12 days of application of Cd were used for measuring slow chlorophyll fluorescence parameters"
Fig. 5c. A letter is missing, indicating reliability - 12 days.
Specify the organ in the Figure 4, 5, 8 legends (root, shoot, whole plant).
L287-290. Text needs editing ("Although" repeat)
Author Response
The authors have done a lot of experimental work and obtained interesting data. However, the conclusions made by the authors, even in the descriptive part, do not correspond well to the data presented in the graphs.
Abstract
- "Peroxisomal Polyamine oxidases (PAOs) PAO2, PAO3, PAO4, and PAO6 expression levels were increased in 12 days following Cd exposure".
This statement is inaccurate. A strong increase only for PAO2 and PAO3, and a transit one. In PAO4 and PAO6, it is almost imperceptible, if compared with their increase at later dates, if at all. That is, the dynamics of these 2 groups is completely different.
Author response: Thank you for your comments, We tried to be more clear in our statement in the introduction.
- "Cd toxicity in days 6 and 15 caused an increase in Ornithine decarboxylase expression, which is involved in putrescine biosynthesis, leading to an elevation in putrescine levels in days 6 and 15." Ornithine decarboxylase transiently increased on day 6 and putrescine only on day 15, which makes the relationship somewhat problematic as both were low on day 12 (at least this delay needs to be explained). In addition, the maximum Cd in the roots is just on the 12th day, and in the shoots on the 15th.
Author response: On days 6 and 15, compared to other time points, the highest level of expression of the ornithine decarboxylase gene and also the level of putrescine in were shown. One of the hypotheses can be that the expression of the ornithine decarboxylase gene increases putrescine and reduces the accumulation of cadmium in the root but why the break on the 12 day is achieved requires further studies. However, what is clear is that the presence of putrescine has reduced the accumulation of cadmium. Since there was no substantiated evidence for this explanation, this hypothesis was not mentioned in the article. However, in the experiment that is being conducted recently, closer time points and the amount of cadmium in all time points are being investigated in order to investigate the cumulative effect of cadmium and its effect on genes as well as its transfer from root to shoot in more detail. Obviously, many questions arise after the end of the experiment and obtaining the results, which can be the basis for conducting further experiments.
- "Spermidine and Spermine levels were reduced on day 6 by Cd application which was parallel with suppressed Spermidine synthase gene. However, an increase in Spermidine and Spermine levels was observed on day 12 along with a significant elevation in Spermidine synthase expression."
According to the Fig. 5b, c, Spermidine and Spermine levels did decrease for a short time on day 6, but already on day 12 this decrease was more than compensated. According to Fig. 6a, no significant decrease in Spermidine synthase gene at day 6 is imperceptible, as well as a significant increase at day 12. The level of the enzyme during the treatment with Cd was slightly reduced throughout the experiment, incl. on 12 days. The most striking detail of the dynamics is the prevention of an increase in the enzyme in the control at day 15, which has little to do with Spermidine and Spermine levels.
Author response: The amount of spermine and spermidine decreased on the 12th day compared to the 6th day for the control treatment (without cadmium), while it increased on the 12th day compared to the 6th day for the cadmium treatment. Besides, in cadmium treatment on day 6, the expression of spermidine synthase showed a significant decrease compared to days 3, 12 and 15, and on day 12, spermine and spermidine content increased in cadmium treatment. It seems that the referee mentioned the control treatments, while we considered the cadmium treatment.
- "On day 6, Cd was observed to start being accumulated in the root and increased expression of microRNA 528; while on day 15, Cd started to be observed in the shooting part which an increase in microRNA 390 and microRNA 168."
Author response: This part was not written correctly in the manuscript, so it was removed according to the reviewer suggestion because the beginning of the presence of cadmium in the stem is on day 6 and the highest amount of cadmium accumulation was observed on day 15.
According to Fig. 2ab, on day 6, Cd began to accumulate in the roots, but in shoots, the “beginning” of accumulation should rather be considered as 12 days, and 15 days is the maximum content of the toxicant. Transient expression of microRNA 528 in experimental plants really corresponds to 6 days, but on day 15, no accumulation of microRNA 390 and microRNA 168 occurred compared to the control.
Author response: In shoot, the accumulation of cadmium started on the 6th day and the highest accumulation was on the 12th and 15th day.
- ''On day 12, the day that the plant launches Cd transport to the shoot, all miRNA expression is reduced."
- - On day 12, the minimum value was observed in the control. There was no decrease in microRNA 528 and microRNA 168 in the experiment compared to the control at all, and for microRNA 390 a slight decrease is most likely just an error in the use of statistics.
Author response: This part was not written correctly in the manuscript, so it was removed according to the reviewer's suggestion
Discussion
- L257-260 Greater cadmium accumulation does not imply greater cadmium sensitivity, and the relationship between cadmium accumulation and high miR390 content does not imply that the mechanism of cadmium toxicity is due to a decrease in miR390 content.
Author response: This interpretation comes from reference number 78 of the manuscript.
- L263-266 " Our data represent that Cd exposure progressively reduces the miR390 level from the first date of application till date 12. Our results also indicate reduced Cd content in root from day 12. Thus, the increase in miR390 from day 12 could be attributable to the Cd translocation from the root to the shoot."
- It follows from Fig. 7a that the level of miR390 is very high in young control plants and drops sharply with age. At the same time, the Cd curve almost completely coincides with the control, except for the starting point (3 days): At points 6, 12 and 20, the level in the Cd variant only slightly fluctuates around the control curve. Since the statistical analysis carried out by the authors confirms significant differences at a number of points, the question arises of how the biological replicates were obtained? Is biological repetition really biological. This can happen if samples for different replications were collected not from the material of individual plants, but from the total sample. If this is the case, then the total variance is underestimated and insignificant differences become significant. When interpreting experimental data, such a danger should be taken into account.
Author response: On the third day, despite the fact that cadmium was not observed in the root, the root system, presumably, has received the signal from the substrate containing cadmium, and it seems likely that Mir390 plays a more effective role in early signaling in the presence of cadmium (according to our observations). Regarding biological repetitions, for each treatment, 3 repetitions, i.e., 3 times of cultivation were done in separate culture trays, and sampling was done from the most available samples of the certain treatment, and samples had maximum similarity in their appearance.
- It seems to me that the following interpretation of the data in Fig. 7a is the most probable. 1) In control, miR390 levels are high in young seedlings and then drop sharply, 2) Treatment dramatically reduces miR390 levels in young seedlings within 3 days post-treatment (miR390 functions are associated with the early period and this is where Cd toxicity appears - however data is limited to one point and require confirmation), 3) after 12 days, miR390 increases slightly regardless of the presence of Cd and its transport.
Author response: This interpretation is correct and to get more insight into the dynamic interaction of cadmium and different MiRNAs, a sets of experiment are in line with closer time point and MiRNAs expression pattern after cadmium treatments.
- L266-267 "In this regard, on day 12, some peroxisomal PAOs (PAO2 and PAO3) levels increased until day 12 followed by a reduction on day 15".
Author response: Since from day 20 still some increase has been shown in some circumstances, therefore ''In this regard, on day 12, peroxisomal PAOs , particularly PAO2 and PAO3 expression levels increased until day 12 followed by a reduction on day 15'' changed to ''In this regard, on day 12, some peroxisomal PAOs (PAO2 and PAO3) levels increased until day 12''.
From fig. 4bc, it can be concluded that this is a transitive increase in 2 out of 4 peroxisomal PAOs.
- L267-268 "In an opposite manner, miR390 expression and PAOs level can represent their negative feed-back regulation under Cd stress."
Author response: We are agree with reviewer and according to the same comment about this interpretation we changed ''Peroxisomal Polyamine oxidases (PAOs) PAO2, PAO3, PAO4, and PAO6 expression levels were increased in 12 days following Cd exposure. Cd toxicity in days 6 and 15 caused an increase in Ornithine decarboxylase expression'' to ''Peroxisomal Polyamine oxidases, particularly (PAOs) PAO2, PAO3, PAO4, and PAO6 expression levels were increased in 12 days following Cd exposure. Cd toxicity in days 6 and 15 caused an increase in Ornithine decarboxylase expression'' in abstract section.
The decrease in miR390 on day 12 occurs independently of Cd, the decrease lasts for several days, in contrast to the transitive increase under the action of Cd in 2 peroxisomal PAOs, i.e., most likely, they are not connected in any way.
Author response: We assume that the referee believes that on day 12, only two peroxisomal PAOs increased and we cannot relate this to the decrease of Mir-390 and the decrease of Mir-390 is independent of cadmium. However we believe that this reduction is significant on the 12th day. Also, on the 12th day, the expression of PAO 5 increased, which includes PAO2, 3, 4, 5, 6 therfore that the sentence written in the manuscript could be considered as correct statement.
- L292-296 "In the current study, Cd caused a reduction in miR168 expression level, as the minimum content was found on day 12, however, a significant increase was observed afterwards. Interlink between an increase in miR168 expression level and Cd exposure is unlikely as its expression level showed an elevation in the control condition as well."
Judging by Fig. 7c, the most interesting moment in the behavior of miR168 is the transit peak on day 6 in the control, i.e. a little later than miR390. However, in both cases, their transit increase is almost completely suppressed by treatment with Cd. So, here the relationship with stress is obvious.
Author response: Authors highly appreciated the reviewer valuable interpretations of the manuscript.
Minor comments
- L54-56 "In addition, Cd can be easily replaced by similar divalent cations in the active sites of antioxidant proteins, leading to 55 an imbalance in cellular reactive oxygen species (ROS)". Please fix the passive form (can easily replace).
Author response: ''Cd can be easily replaced by similar divalent cations in the active sites of antioxidant proteins'' Changed to '' Cd can easily replace by similar divalent cations in the active sites of antioxidant proteins''
- L58-59 "The long half-life of Cd (10 to 30 years) raises concern about its accumulation in the body and its risk to human health [5]". Specify what half-life you are talking about: Cd is not a radioactive element.
Author response: this sentence was removed according to the suggestion of one of the other reviewers as well.
Methods
- How many plants are in each replication?
Author response: 30 plants, it is now indicated in the revised manuscript
- 3. Cd determination Indicate the weight of the sample and how many plants the tissue for analysis was taken from.
Author response: The number of plants (root or shoot) was selected according to the time point. For earlier time point (3 and 6 days after Cd exposure) up to 8 plants (root or shoot) and for later time point (12, 15 and 20 days after Cd exposure) up to 4 plants (root or shoot) were selected.
- 2 What is the reason for the simultaneous decrease in the concentration of cadmium in the roots and shoots on the 20th day?
Author response: It seems that on the 20th day, the root and aerial parts of the plant become larger, and as a result, the amount of cadmium per kilogram of the plant shows a reduction.
- 3. Specify the measurement time. In the methodology: "4, 8 and 12 days of application of Cd were used for measuring slow chlorophyll fluorescence parameters"
Author response: This experiment was conducted 20 days after Cd exposure and it has been included in the figure legend and '' 5.4. Transient and slow induction of chlorophyll fluorescence'' part in section method.
- 5c. A letter is missing, indicating reliability - 12 days.
Specify the organ in the Figure 4, 5, 8 legends (root, shoot, whole plant).
Author response: Legends were corrected according to the reviewer's suggestion.
- L287-290. Text needs editing ("Although" repeat)
Author response: Revision was done according to the reviewer's suggestion.
Round 2
Reviewer 5 Report
Unfortunately, the revised version contains almost all the shortcomings of the previous one. The description of the experiments and the conclusions for the most part do not correspond to the results obtained and presented in the figures.
Perhaps some old version of the manuscript was uploaded. I didn't find any changes to the microRNA part that were declared in the responses.
Abstract
1. "Peroxisomal Polyamine oxidases (PAOs) PAO2, PAO3, PAO4, and PAO6 expression levels were increased in 12 days following Cd exposure".
This statement is inaccurate. A strong increase only for PAO2 and PAO3, and a transit one. In PAO4 and PAO6, it is almost imperceptible, if compared with their increase at later dates, if at all. That is, the dynamics of these 2 groups is completely different.
Author response: Thank you for your comments, We tried to be more clear in our statement in the introduction.
Note:
The essence of the research should be reflected in the Abstract, regardless of the Introduction section. And this phrase in the abstract, as before, does not reflect the data obtained by the authors, in particular, presented in Fig.4. Formally, there is a difference with control, but only formally. Due to the difference in the dynamics of different PAOs, this phrase does not reflect the essence of the processes observed by the authors and has no scientific meaning. For example, in the case of PAO4, inhibition dominates at a later date, and a very small, although formally significant increase on day 12, if not an error, is unlikely to play a role in the plant's response to stress.
2. "Cd toxicity in days 6 and 15 caused an increase in Ornithine decarboxylase expression, which is involved in putrescine biosynthesis, leading to an elevation in putrescine levels in days 6 and 15." Ornithine decarboxylase transiently increased on day 6 and putrescine only on day 15, which makes the relationship somewhat problematic as both were low on day 12 (at least this delay needs to be explained). In addition, the maximum Cd in the roots is just on the 12th day, and in the shoots on the 15th.
Author response: On days 6 and 15, compared to other time points, the highest level of expression of the ornithine decarboxylase gene and also the level of putrescine in were shown. One of the hypotheses can be that the expression of the ornithine decarboxylase gene increases putrescine and reduces the accumulation of cadmium in the root but why the break on the 12 day is achieved requires further studies. However, what is clear is that the presence of putrescine has reduced the accumulation of cadmium. Since there was no substantiated evidence for this explanation, this hypothesis was not mentioned in the article. However, in the experiment that is being conducted recently, closer time points and the amount of cadmium in all time points are being investigated in order to investigate the cumulative effect of cadmium and its effect on genes as well as its transfer from root to shoot in more detail. Obviously, many questions arise after the end of the experiment and obtaining the results, which can be the basis for conducting further experiments.
Note:
The question was not about the need for clarifying experiments, but about the lack of an adequate description of those already carried out. Figure 6b shows a single transient peak of ornithine decarboxylase on day 6 (with a statistically significant aftereffect on days 12, 15, and 20), and this, quite possibly, caused the appearance again of a single peak of putrescine content at day 15 day according to Fig.5a.
3. "Spermidine and Spermine levels were reduced on day 6 by Cd application which was parallel with suppressed Spermidine synthase gene. However, an increase in Spermidine and Spermine levels was observed on day 12 along with a significant elevation in Spermidine synthase expression."
According to the Fig. 5b, c, Spermidine and Spermine levels did decrease for a short time on day 6, but already on day 12 this decrease was more than compensated. According to Fig. 6a, no significant decrease in Spermidine synthase gene at day 6 is imperceptible, as well as a significant increase at day 12. The level of the enzyme during the treatment with Cd was slightly reduced throughout the experiment, incl. on 12 days. The most striking detail of the dynamics is the prevention of an increase in the enzyme in the control at day 15, which has little to do with Spermidine and Spermine levels.
Author response: The amount of spermine and spermidine decreased on the 12th day compared to the 6th day for the control treatment (without cadmium), while it increased on the 12th day compared to the 6th day for the cadmium treatment. Besides, in cadmium treatment on day 6, the expression of spermidine synthase showed a significant decrease compared to days 3, 12 and 15, and on day 12, spermine and spermidine content increased in cadmium treatment. It seems that the referee mentioned the control treatments, while we considered the cadmium treatment.
Note:
The dynamics of Spermidine and Spermine levels in Fig. 5, in my opinion, is as follows: no change compared to control on day 3, decrease compared to control on day 6, increase compared to control on day 12, no change compared to control on the 15th day. There is nothing even close to similar in the dynamics of the expression of spermidine synthase: the expression level during Cd treatment is stable and significantly lower than the control level throughout the entire experiment. The study of stress exposure involves comparison with control. That is, "parallel in experiments with stress exposure" - this means that the increase compared to the control of any substance with a certain lag period follows the increase compared to the control of its enzyme. If the authors are interested in the dynamics of indicators within a separate variant of Cd treatment, then again this should be compared with the dynamics within the control.
Author Response
Thank you for providing us with a chance to revise our manuscript. Following this letter is a point-wise response to the reviewer’s comments. Changes made in the manuscript are marked using track changes. The revision has been developed in consultation with all coauthors, and each author has given approval to the final form of this revision.
- This statement is inaccurate. A strong increase only for PAO2 and PAO3, and a transit one. In PAO4 and PAO6, it is almost imperceptible, if compared with their increase at later dates, if at all. That is, the dynamics of these 2 groups is completely different.
Author response: Thank you for your comments, we tried to be clearer in our statement in the introduction.
Note:
The essence of the research should be reflected in the Abstract, regardless of the Introduction section. And this phrase in the abstract, as before, does not reflect the data obtained by the authors, in particular, presented in Fig.4. Formally, there is a difference with control, but only formally. Due to the difference in the dynamics of different PAOs, this phrase does not reflect the essence of the processes observed by the authors and has no scientific meaning. For example, in the case of PAO4, inhibition dominates at a later date, and a very small, although formally significant increase on day 12, if not an error, is unlikely to play a role in the plant's response to stress.
Author response: Thank you for these observations. We have rewritten the abstract to better differentiate among the objectives so that we deleted the sentence of "Peroxisomal Polyamine oxidases (PAOs) PAO2, PAO3, PAO4, and PAO6 expression levels were increased in 12 days following Cd exposure" as we are agree with the reviewer (L31-L35), however to get better insight into the changes in PAOs pattern we added more discussion according to the reviewer comment in discussion section (L342-L355).
- Cd toxicity in days 6 and 15 caused an increase in Ornithine decarboxylase expression, which is involved in putrescine biosynthesis, leading to an elevation in putrescine levels in days 6 and 15." Ornithine decarboxylase transiently increased on day 6 and putrescine only on day 15, which makes the relationship somewhat problematic as both were low on day 12 (at least this delay needs to be explained). In addition, the maximum Cd in the roots is just on the 12th day, and in the shoots on the 15th.
Author response: On days 6 and 15, compared to other time points, the highest level of expression of the ornithine decarboxylase gene and also the level of putrescine in were shown. One of the hypotheses can be that the expression of the ornithine decarboxylase gene increases putrescine and reduces the accumulation of cadmium in the root but why the break on the 12 day is achieved requires further studies. However, what is clear is that the presence of putrescine has reduced the accumulation of cadmium. Since there was no substantiated evidence for this explanation, this hypothesis was not mentioned in the article. However, in the experiment that is being conducted recently, closer time points and the amount of cadmium in all time points are being investigated in order to investigate the cumulative effect of cadmium and its effect on genes as well as its transfer from root to shoot in more detail. Obviously, many questions arise after the end of the experiment and obtaining the results, which can be the basis for conducting further experiments.
Note:
The question was not about the need for clarifying experiments, but about the lack of an adequate description of those already carried out. Figure 6b shows a single transient peak of ornithine decarboxylase on day 6 (with a statistically significant aftereffect on days 12, 15, and 20), and this, quite possibly, caused the appearance again of a single peak of putrescine content at day 15 day according to Fig.5a.
Author response: We agree with the reviewer and "Cd toxicity in days 6 and 15 caused an increase in Ornithine decarboxylase expression, which is involved in putrescine biosynthesis, leading to an elevation in putrescine levels in days 6 and 15." was changed to ''The expression level of the ornithine decarboxylase (ORDC) increased from the day 6 under Cd exposure compared to the control. However, Cd toxicity only on day 15 led to an increase in putrescine (Put) content compared to the control. In fact, with the exception of day 15, the increase in ORDC transcript levels does not show a direct correlation with the observed increase in Put content.'' in abstract section. Moreover, we discussed this statement in the manuscripts as marked using track changes in discussion part (L398-L404).
- "Spermidine and Spermine levels were reduced on day 6 by Cd application which was parallel with suppressed Spermidine synthase gene. However, an increase in Spermidine and Spermine levels was observed on day 12 along with a significant elevation in Spermidine synthase expression."
According to the Fig. 5b, c, Spermidine and Spermine levels did decrease for a short time on day 6, but already on day 12 this decrease was more than compensated. According to Fig. 6a, no significant decrease in Spermidine synthase gene at day 6 is imperceptible, as well as a significant increase at day 12. The level of the enzyme during the treatment with Cd was slightly reduced throughout the experiment, incl. on 12 days. The most striking detail of the dynamics is the prevention of an increase in the enzyme in the control at day 15, which has little to do with Spermidine and Spermine levels.
Author response: The amount of spermine and spermidine decreased on the 12th day compared to the 6th day for the control treatment (without cadmium), while it increased on the 12th day compared to the 6th day for the cadmium treatment. Besides, in cadmium treatment on day 6, the expression of spermidine synthase showed a significant decrease compared to days 3, 12 and 15, and on day 12, spermine and spermidine content increased in cadmium treatment. It seems that the referee mentioned the control treatments, while we considered the cadmium treatment.
Note:
The dynamics of Spermidine and Spermine levels in Fig. 5, in my opinion, is as follows: no change compared to control on day 3, decrease compared to control on day 6, increase compared to control on day 12, no change compared to control on the 15th day. There is nothing even close to similar in the dynamics of the expression of spermidine synthase: the expression level during Cd treatment is stable and significantly lower than the control level throughout the entire experiment. The study of stress exposure involves comparison with control. That is, "parallel in experiments with stress exposure" - this means that the increase compared to the control of any substance with a certain lag period follows the increase compared to the control of its enzyme. If the authors are interested in the dynamics of indicators within a separate variant of Cd treatment, then again this should be compared with the dynamics within the control.
Author response: We believe that except for day 20, the pattern of changes in spermine and spermidine content looks the same in Fig 5 either compare with the control or individually. This trend is consistent with the expression of spermidine synthase in Fig 6, which decreased on the day 6 and increased on the day 12 (under Cd stress). The fact that the change in spermidine content is not close to the level of gene expression can be due to the presence of introns in this gene, which indicates post-transcriptional regulation and alternative splicing, and as a result the presence of different transcripts [1]. Rodríguez-Kessler and colleagues revealed that Zmspds2A and Zmspds2B are generated by alternative splicing in Zea mays. In fact, the increase in spermidine content on day 12 in control plants can be the result of the increase in other spermidine transcripts expression that are generated by the gene splicing. Because it has been shown in previous studies that the genes responsible for spermine and spermidine biosynthesis have a large number of introns that are capable of synthesizing various alternatives leading to different transcript, though the final products can be the same [2,3] This could help plant's flexibility in biosynthesis of PAs against dynamic environmental stimuli.
- Rodríguez-Kessler, M.; Alpuche-Solís, A.G.; Ruiz, O.A.; Jimenez-Bremont, J.F. Effect of salt stress on the regulation of maize (Zea mays L.) genes involved in polyamine biosynthesis. Plant Growth Regul 2006, 48, 175-185.
- Panicot, M.; Minguet, E.G.; Ferrando, A.; Alcázar, R.; Blázquez, M.A.; Carbonell, J.; Altabella, T.; Koncz, C.; Tiburcio, A.F. A polyamine metabolon involving aminopropyl transferase complexes in Arabidopsis. The Plant Cell 2002, 14, 2539-2551.
- Hanzawa, Y.; Imai, A.; Michael, A.J.; Komeda, Y.; Takahashi, T. Characterization of the spermidine synthase-related gene family in Arabidopsis thaliana. FEBS Lett 2002, 527, 176-180.
Round 3
Reviewer 5 Report
Author response: 1) We believe that except for day 20, the pattern of changes in spermine and spermidine content looks the same in Fig 5 either compare with the control or individually. 2) This trend is consistent with the expression of spermidine synthase in Fig 6, which decreased on the day 6 and increased on the day 12 (under Cd stress). The fact that the change in spermidine content is not close to the level of gene expression can be due to the presence of introns in this gene, which indicates post-transcriptional regulation and alternative splicing, and as a result the presence of different transcripts [1]. Rodríguez-Kessler and colleagues revealed that Zmspds2A and Zmspds2B are generated by alternative splicing in Zea mays. In fact, the increase in spermidine content on day 12 in control plants can be the result of the increase in other spermidine transcripts expression that are generated by the gene splicing. Because it has been shown in previous studies that the genes responsible for spermine and spermidine biosynthesis have a large number of introns that are capable of synthesizing various alternatives leading to different transcript, though the final products can be the same [2,3] This could help plant's flexibility in biosynthesis of PAs against dynamic environmental stimuli.
Note 2: Text (1) is quite consistent with Fig 5. Text (2) is probably also correct under Cd stress on its own. However, the range of these changes (1 and 2) looks completely insignificant compared to the inhibition by cadmium of SPDS transient expression in the control on day 15. And this is probably just one of the interesting results that requires discussion, but is not even mentioned as important. In the control variant, there is no correlation between Spermidine and Spermine and SPDS. Why are subtle associations of Spermidine and Spermine with SPDS under Cd stress mentioned in the Abstract, but the absolute absence of such a correlation in the control and the inhibition of SPDS itself is not?